# DIFFERENTIALLY PRIVATE SYNTHETIC DATA VIA APIS 4: TABULAR DATA

## ABSTRACT

Tabular data is one of the most widely used formats in practice, yet much of it remains inaccessible due to privacy concerns. Synthetic data generation with formal privacy guarantees, i.e. differential privacy (DP), offers a promising solution to enable data sharing while protecting sensitive information. Despite extensive study, state-of-the-art methods often focus on minimizing low-order marginal query errors and overlook the challenges posed by high-order correlations. To address this gap, we adapt the Private Evolution (PE) framework, originally developed for DP-compliant image and text synthesis, to tabular data. We introduce Tab-PE – an algorithm for generating synthetic tabular data under DP. Tab-PE refines a synthetic dataset by an evolutionary process that leverages APIs to generate variations of the data, privately evaluate them, and retain the highest-quality samples. While the original PE requires access to large foundation models, Tab-PE is computationally efficient with heuristic APIs specialized for tabular data. Through extensive experiments on real-world and simulation datasets, we demonstrate that Tab-PE substantially outperforms prior baselines on datasets exhibiting high-order correlations. Compared to the best baseline – AIM, Tab-PE improves classification accuracy by up to 10% while running $28\times$ faster.

## 1 INTRODUCTION

Tabular data is an important type of data that is widely used in many domains. However, because it often contains sensitive information such as patient records and financial transactions, using and sharing such data are challenging due to potential risks of exposing private information (Borisov et al., 2024). To tackle the privacy concerns, generating synthetic tabular data with differential privacy (DP) guarantees has been a long-standing and active research area (Li et al., 2014; Zhang et al., 2021; Liu et al., 2021; McKenna et al., 2022; Liu et al., 2023; Tran & Xiong, 2024; Cormode et al., 2025). This synthetic data can be used for various purposes – such as data analysis, machine learning model training, and sharing with third parties – while still providing formal privacy guarantees for individual records in the original dataset.

Despite this promise, generating realistic tabular data remains challenging due to difficulties in capturing complex multi-dimensional data distributions under the privacy constraints. State-of-the-art (SOTA) methods (McKenna et al., 2022; Liu et al., 2021; 2023) address this by estimating low-order statistical queries (typically marginals) and then stitching them together to approximate the full data distribution. However, these methods have a fundamental limitation: they do not scale well to model high-order correlations as the number of queries grows exponentially with the order (i.e., the number of involved attributes). Since DP requires adding noise to each query answer, the noise accumulates as the number of queries increases. Therefore, estimating a large number of queries under strict privacy constraints is challenging and often leads to low-quality measurements.

***Most prior evaluations sidestep the challenge of high-order correlations***. Popular datasets used in the literature appear to be dominated by low-order dependencies (Chen et al., 2025; Tao et al., 2022). Intuitively, we measure the order of correlations in a dataset by considering the downstream performance gap of simple classifiers that capture only low-order correlations (e.g., shallow decision trees) versus complex classifiers that leverage high-order correlations (e.g., deep trees). When the performance gap between these two types of classifiers is small, the dataset primarily reflects low-order correlations. Indeed, many commonly used datasets such as Adult, Bank, and Census have

this property. Varying the maximum depth of the XGBoost trees (Chen & Guestrin, 2016) yields trivial performance differences (typically <1%) (Fig. 7, App. B.1). This characteristic makes the existing leading methods using statistical queries appear highly effective, even though they do not model high-order correlations. Consequently, much of the field has been implicitly optimized for these favorable settings, while leaving open the question of whether the current methods can truly preserve high-order correlations that are not revealed by standard benchmarks.

In this work, we focus on investigating this gap. We construct a stress test with XOR correlations and show that SOTA methods quickly fail to capture such high-order correlations (Fig. 1). To address this challenge, we propose a method based on the Private Evolution (PE) framework (Lin et al., 2024), tailored for tabular data – named Tab-PE. PE is a breakthrough that has shown promising results in generating high-quality synthetic data in other domains such as images (Lin et al., 2024; 2025) and texts (Xie et al., 2024; Hou et al., 2024; 2025; Wang et al., 2025). It generates synthetic data through an iterative process of generating variations of the data and then selecting the best ones based on a DP voting mechanism. Previous methods have designed APIs for generating variations of images or texts such as using foundation models (Lin et al., 2024; Xie et al., 2024; Wang et al., 2025) or using simulators (Lin et al., 2025). For tabular data, Swanberg et al. (2025) argue Private Evolution with API access to LLMs does not perform satisfactorily.

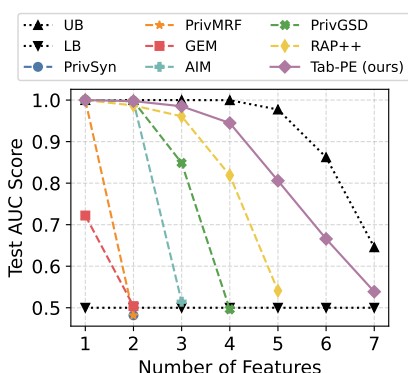

Figure 1: Stress test for high-order correlation modeling with XOR simulation datasets at $\epsilon = 1.0$. UB stands for Upper Bound using private data ($\epsilon = \infty$). LB presents random guess performance.

We design simple yet effective and efficient APIs for generating variations of tabular data *without using any foundation models*. Building on the PE framework, Tab-PE first initializes a random synthetic dataset then iteratively refines it. In each iteration, we generate variations by simply adding controlled random noise to numerical features and resampling categorical features with a scheduled probability. The synthetic samples are then scored by a DP voting mechanism based on full-record nearest-neighbor matching to private data, which can implicitly capture complex, high-dimensional dependencies. High-scoring samples are selected for the next iteration, enabling an iterative refinement process. We show that Tab-PE outperforms SOTA methods on a wide range of settings and is the most computationally efficient. Overall, our contributions can be summarized as follows:

- We revisit the challenge of modeling high-order correlations in differentially private synthetic tabular data generation. Our stress test reveals that SOTA methods fail to capture such correlations.

- We propose Tab-PE, a method based on the Private Evolution framework, with simple yet effective and efficient APIs for generating variations of tabular data without using any foundation models.

- We conduct extensive experiments on a broad collection of new datasets and settings, going beyond the standard benchmarks that mainly reflect low-order correlations. Our results indicate that Tab-PE consistently outperforms the baselines, especially under strict privacy regimes. Tab-PE is also the most computationally efficient method and faster than utility-competitive baselines up to $30\times$ without requiring GPUs.

## 2 RELATED WORKS

**Differentially Private Tabular Synthesis**. DP synthetic tabular data is a long-standing problem with many prior works (Yang et al., 2024; Cormode et al., 2025). In a real-world competition (NIST, 2018), the winning solutions are dominated by methods that rely on marginal queries such as MST (McKenna et al., 2021), DPSyn (Li et al., 2021), and PrivBayes Zhang et al. (2017). All these methods first answer the low-order marginal queries in a DP manner, then reconstruct the synthetic data from the noisy answers with different techniques, e.g., probabilistic graphical

models (PGMs) (McKenna et al., 2019) and Bayesian networks. To improve this pipeline, more advanced methods (AIM (McKenna et al., 2022), MRF (Cai et al., 2021)) dynamically select suitable marginal queries. Subsequently, RAP (Liu et al., 2021), RAP++ (Vietri et al., 2022), PrivGSD (Liu et al., 2023), and PrivPGD (Donhauser et al., 2024) consider generation as an optimization process that iteratively refines the synthetic dataset to minimize the error on the noisy answers. Meanwhile, JAM (Fuentes et al., 2024) aims to utilize publicly available data. Beyond the methods using statistical queries, there is a line of research that leverages machine learning for this problem. Inspired by the success of image generation, some works employ GANs (Xie et al., 2018; Yoon et al., 2019). However, it turns out that GAN-based methods do not align well with DP noise due to its complex architecture and adversarial training process (Cormode et al., 2025). Some recent works explore transformer-based architectures (Castellon et al., 2023; Sablayrolles et al., 2023), and large-language models (Tran & Xiong, 2024). Although the gap between these and the marginal-based methods is smaller than GANs, they still lag behind the marginal-based methods. A recent benchmark (Chen et al., 2025) confirms that the marginal-based methods still dominate the field. In this work, we revisit the problem with a perspective of high-order correlations and propose a new efficient and effective framework that does not rely on statistical queries or model training.

**Private Evolution**. PE is a breakthrough for synthetic data generation with DP. PE was first introduced by Lin et al. (2024) for images. Unlike previous synthesizers, which require model training/fine-tuning on private data (Kurakin et al., 2024; Dockhorn et al., 2023), PE instead leverages API access to pretrained foundation models. By employing an evolutionary process that iteratively refines the synthetic data, PE achieves SOTA results while being computationally efficient. Xie et al. (2024) extended PE to text, demonstrating its effectiveness by significantly outperforming LLM DP fine-tuning baselines. Zou et al. (2025) enhanced the performance for text by utilizing multiple LLMs via a weighted fusion mechanism. Moreover, the PE framework has been adapted to federated learning settings to reduce communication costs while achieving better utility for language modeling (Hou et al., 2024; 2025). While most PE-based works rely on foundation models, Lin et al. (2025) showed that PE can also be applied to simulators. Additionally, Zhang et al. (2025) modified PE for few-shot generation, while González et al. (2025) studied theoretical convergence aspects of PE. For tabular data, Swanberg et al. (2025) applied PE with LLM-guided APIs. However, the authors argue that PE with LLM API access does not perform satisfactorily. While our work does not contradict their message, we demonstrate that PE using heuristic APIs (without any foundation models) and appropriate designs can be both effective and computationally efficient.

## 3 METHODOLOGY – TAB-PE

**Differential Privacy**. $(\epsilon, \delta)$-differential privacy (DP) is a property of a randomized algorithm $\mathcal{M}$ that guarantees that the output of $\mathcal{M}$ does not change much whether we add or remove any particular entry in the input. More precisely, given any two neighboring datasets $\mathcal{D}, \mathcal{D}'$ (one can be obtained from the other by deleting a single entry) and any possible set of outputs $S$, it holds that $\Pr[\mathcal{M}(\mathcal{D}) \in S] \leq e^\epsilon \Pr[\mathcal{M}(\mathcal{D}') \in S] + \delta$ (Dwork et al., 2014).

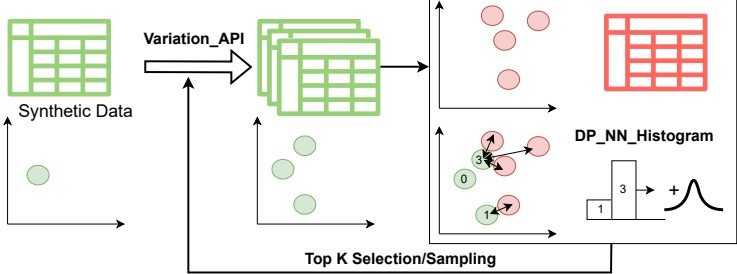

Figure 2: Illustration of Tab-PE. The process starts with an initial set of synthetic samples and iteratively refines them through variations and private scoring.

**Overview**. Let $s$ be a sample $s = \{x_{\text{cat}(1)}, x_{\text{cat}(2)}, ..., x_{\text{num}(1)}, x_{\text{num}(2)}, ..., c\}$, where $x_{\text{cat}}$ denotes categorical attributes, $x_{\text{num}}$ refers to numerical attributes, and $c$ is the class label. $\mathcal{X}_{(i)}$ is the domain of attribute $i$. Given a private dataset $\mathcal{D}_{\text{priv}}$ of samples, our goal is to generate a synthetic dataset $\mathcal{D}_{\text{syn}}$

that preserves the statistical properties of $\mathcal{D}_{\text{priv}}$ while ensuring DP. Our approach, Tab-PE, consists of three main components: (1) a RANDOM_API that generates an initial set of synthetic samples, (2) a VARIATION_API that creates variations of existing samples to explore the sample space, and (3) a DP_NN_HISTOGRAM function that scores each sample in a DP manner. The overall process is an evolutionary loop, illustrated in Fig. 2. For each iteration, we generate variations of the current samples using VARIATION_API, evaluate them using the private dataset with DP_NN_HISTOGRAM, and retain the top-scoring samples to form the next generation. This process continues for a predefined number of iterations or until convergence.

**RANDOM_API**. The RANDOM_API generates an initial set of the synthetic samples. For categorical attribute $x_{\text{cat}(i)}$, it randomly selects a value from the set of possible categories $\mathcal{X}_{\text{cat}(i)}$. For numerical attribute $x_{\text{num}(j)}$, it uniformly samples values within the range $(\min_{\mathcal{X}_{\text{num}(j)}}, \max_{\mathcal{X}_{\text{num}(j)}})$. This ensures that the initial synthetic samples are diverse and cover the attribute space.

$$
\begin{aligned}
&\text{RANDOM\_API}(n) = \{s_1, s_2, \ldots, s_n\} \text{ where} \\
&\quad s_k = \{x_{\text{cat}(1)}, x_{\text{cat}(2)}, \ldots, x_{\text{num}(1)}, x_{\text{num}(2)}, \ldots, c\}, \\
&\quad x_{\text{cat}(i)} \sim \text{Uniform}(\mathcal{X}_{\text{cat}(i)}), x_{\text{num}(j)} \sim \text{Uniform}(\min(\mathcal{X}_{\text{num}(j)}), \max(\mathcal{X}_{\text{num}(j)}))
\end{aligned}
\tag{1}
$$

**VARIATION_API**. The API generates perturbed variations of existing samples to explore the sample space. The variation degree $m$ is the number of variations generated per sample. We implement a simple but effective ***random walk*** strategy. For a categorical attribute $x_{\text{cat}(i)} \in \mathcal{X}_{\text{cat}(i)}$, the variation is produced by resampling from its domain with a controlled categorical mutation rate $\mu_{\text{cat}} \in [0, 1]$.

$$
x'_{\text{cat}(i)} \sim
\begin{cases}
x_{\text{cat}(i)}, & \text{with probability } 1 - \mu_{\text{cat}}, \\
\text{Uniform}(\mathcal{X}_{\text{cat}(i)}), & \text{with probability } \mu_{\text{cat}}.
\end{cases}
\tag{2}
$$

For a numerical attribute $x_{\text{num}(j)} \in \mathcal{X}_{\text{num}(j)}$, the variation is generated by adding controlled Gaussian perturbation with scale controlled by a numerical mutation rate $\mu_{\text{num}} \in [0, 1]$ and projecting back into the valid range:

$$
x'_{\text{num}(j)} = \Pi_{\mathcal{X}_{\text{num}(j)}}(x_{\text{num}(j)} + \phi), \quad \phi \sim \mathcal{N}(0, \sigma^2), \sigma = \mu_{\text{num}} \cdot (\max(\mathcal{X}_{\text{num}(j)}) - \min(\mathcal{X}_{\text{num}(j)}))
\tag{3}
$$

where $\Pi_{\mathcal{X}}(\cdot)$ denotes projection onto the feasible range of $\mathcal{X}_{\text{num}(j)}$[1].

Both mutation rates $\mu_{\text{cat}}$ and $\mu_{\text{num}}$ follow a polynomial decay as a function of the iteration index $t$ to balance exploration and exploitation. In the early stages, higher mutation rates encourage exploration of the sample space, while in later stages, lower rates focus on refining high-quality samples.

$$
\mu = \mu_{\text{init}} - (\mu_{\text{init}} - \mu_{\text{final}}) \cdot (t/T)^\gamma
\tag{4}
$$

**DP_NN_HISTOGRAM**. The DP_NN_HISTOGRAM scores synthetic samples in a DP manner. At each iteration $t$, Tab-PE maintains a population $P$, which is a set of candidate synthetic samples, mainly generated by VARIATION_API. We denote a histogram $hist$, where each bin $hist[i]$ corresponds to a sample $P[i]$ in $P$. The value $hist[i]$ represents the count of private samples in $\mathcal{D}_{\text{priv}}$ whose nearest neighbor in $P$ is $P[i]$. The pseudocode of DP_NN_HISTOGRAM is presented in Algo. 1. For each sample in the private dataset $\mathcal{D}_{\text{priv}}$, we find its nearest neighbor in $P$ and increment the corresponding bin (Algo. 1, Lines 2– 4). To ensure DP, we add Gaussian noise to each bin of the histogram (Algo. 1, Line 6). As each private sample can only affect one bin, the

---

**Algorithm 1:** DP_NN_HISTOGRAM

**Input:** Private dataset $\mathcal{D}_{\text{priv}}$, Population $P$,
      Noise multiplier $\sigma$
**Output:** Noisy histogram $hist$
1   $hist \leftarrow [0, 0, ..., 0]$
2   **for** *each sample* $s \in \mathcal{D}_{\text{priv}}$ **do**
3      $\text{i} \leftarrow \text{argmin}_j \text{ distance}(s, P[j])$
4      $hist[\text{i}] \leftarrow hist[\text{i}] + 1$
5   **for** *each index* $i$ *in* $hist$ **do**
6      $hist[\text{i}] \leftarrow hist[\text{i}] + \mathcal{N}(0, \sigma^2)$
7   **return** $hist$

---

sensitivity of this histogram query is 1. By adding noise drawn from $\mathcal{N}(0, \sigma^2)$ to each bin, we achieve $(\epsilon, \delta)$-DP, where $\epsilon$ and $\delta$ are determined by the noise multiplier $\sigma$ and the number of iterations $T$. The privacy analysis can be reused from the Gaussian mechanism and the composition

---

[1]Numerical bounds are assumed known, as the default setting of a widely used library (Holohan et al., 2019)

theorem, as done in the original private evolution paper Lin et al. (2024) and detailed in App. A.1. The distance metric between samples is the mixed-type distance defined as follows, where $\lambda$ is a hyperparameter to balance the contributions of categorical and numerical attributes.

$$\text{distance}(s_a, s_b) = \sqrt{\lambda \sum_i \mathbb{1}\left(x_{\text{cat}(i)}^{(a)} \neq x_{\text{cat}(i)}^{(b)}\right) + \sum_j \left(\frac{x_{\text{num}(j)}^{(a)} - x_{\text{num}(j)}^{(b)}}{\max_{\mathcal{X}_{\text{num}(j)}} - \min_{\mathcal{X}_{\text{num}(j)}}}\right)^2} \tag{5}$$

**Tabular Private Evolution**. The overall process of Tab-PE is summarized in Algo. 2. We first initialize a synthetic dataset $\mathcal{D}_{\text{syn}}$ with RANDOM_API. Then we iteratively refine the synthetic samples over $T$ iterations. In each iteration, we generate a population of sample candidates using VARIATION_API, score them with DP_NN_HISTOGRAM, and select the top samples to form the next generation. To enhance exploration and exploitation, we employ a two-stage approach: sampling with replacement in the early iterations, followed by ranking and selecting the top samples in later iterations. In the first $T_{\text{sampling}}$ iterations, we sample new synthetic samples based on the noisy histogram-based probabilities. The variation degree $m$ is set to 1 (Algo. 2, Line 8) to maintain a small population size, which yields higher average histogram counts (Algo. 1, Lines 2–4) and thus reduces sensitivity to noise (Algo. 1, Line 6). This leads to more reliable sampling probabilities (Algo. 2, Line 14). In the second stage, we set $m$ to a higher value to encourage local refinement. The population $P$ now includes both the variations and the previous selected samples (Algo. 2, Line 10). We then select the top $N^{(c)}$ samples based on their noisy histogram scores (Algo. 2, Line 17). Intuitively, at the beginning, some samples may have significantly large counts and sampling with replacement allows these samples to be selected multiple times, which helps to quickly shift the distribution of synthetic samples towards the private data distribution. In the later stage, selecting the top samples helps to locally refine the synthetic dataset and improve its quality. This two-stage approach effectively exploits the strengths of both sampling and top selection, leading to better overall performance.

---

**Algorithm 2:** Tabular Private Evolution

**Input:** The set of classes $C$, Private dataset $\mathcal{D}_{\text{priv}}$, Noise multiplier $\sigma$,
      Number of iterations $T$, Number of sampling iterations $T_{\text{sampling}}$,
      Variation degree $m$, Number of synthetic samples $N$

**Output:** Synthetic dataset $\mathcal{D}_{\text{syn}}$

1  $\mathcal{D}_{\text{syn}} \leftarrow \emptyset$
2  **for** *each class* $c \in C$ **do**
3      $\mathcal{D}_{\text{priv}}^{(c)} \leftarrow$ subset of $\mathcal{D}_{\text{priv}}$ of class $c$
4      $N^{(c)} \leftarrow N \cdot |\mathcal{D}_{\text{priv}}^{(c)}|/|\mathcal{D}_{\text{priv}}|^2$                 /*Num synthetic samples of class $c$*/
5      $\mathcal{D}_0 \leftarrow$ RANDOM_API($N^{(c)}$);                      /*Initialize a dataset*/
6      **for** $t \leftarrow 1$ **to** $T$ **do**
7          **if** $t \leq T_{\text{sampling}}$ **then**
8              $P_t \leftarrow$ VARIATION_API($\mathcal{D}_{t-1}$, 1)              /*Population at $t$*/
9          **else**
10             $P_t \leftarrow$ VARIATION_API($\mathcal{D}_{t-1}$, $m$) $\cup \mathcal{D}_{t-1}$     /*Population at $t$*/
11          $hist_t \leftarrow$ DP_NN_HISTOGRAM($\mathcal{D}_{\text{priv}}^{(c)}$, $P_t$, $\sigma$)
12         **if** $t \leq T_{\text{sampling}}$ **then**
13              $hist_t[i] \leftarrow \max(0, hist_t[i])$        /*Clamp negative counts to zero*/
14              $prob[i] \leftarrow hist_t[i]/\sum_j hist_t[j]$
15              $\mathcal{D}_t \leftarrow$ sample $N^{(c)}$ samples from $P_t$ with replacement according to $prob$
16         **else**
17              $\mathcal{D}_t \leftarrow$ top $N^{(c)}$ samples of $P_t$ by $hist_t$
18      $\mathcal{D}_{\text{syn}} \leftarrow \mathcal{D}_{\text{syn}} \cup \mathcal{D}_T$
19  **return** $\mathcal{D}_{syn}$

---

The previous query-based methods require answering many queries. Each single query needs to scan the entire dataset. Moreover, high-dimensional queries involving many attributes are especially

---

[2]We assume class distributions are known as (Lin et al., 2024; Xie et al., 2024). Additionally, experiments in App. C.8 show Tab-PE performs similarly either w/ or w/o this assumption.

costly, as they create large multi-way count tables that consume significant memory and computation resources. Additionally, model fitting and optimization over these query measurements usually requires iterative solvers that may scale poorly with the dimensionality. In contrast, Tab-PE operates at the sample level, and each iteration only requires a single pass over the private dataset to conduct nearest neighbor search. While the query-based methods struggle to handle high-order correlations due to the exponential growth of queries, Tab-PE leverages full-record nearest neighbor matching and iterative refinement that can implicitly capture complex, high-dimensional dependencies.

## 4 EXPERIMENTS AND RESULTS

**Overview**. As we focus on high-order correlation modeling in DP tabular data, we first examine the algorithmic capability of the baselines and Tab-PE by an extreme case of XOR simulation datasets. We then conduct extensive experiments on realistic simulated datasets with multiple non-linear underlying functions and real-world datasets with high-order correlations, under various privacy constraint settings. We also evaluate the methods on widely-used real-world datasets with predominantly low-order correlations. Finally, we examine computational efficiency and analyze the technical design choices in Tab-PE.

### 4.1 EXPERIMENT SETUP

**Baselines**. We consider several SOTA baselines in DP tabular data synthesizers, following a recent benchmark (Chen et al., 2025): PrivSyn (Zhang et al., 2021), PrivMRF (Cai et al., 2021), GEM (Vietri et al., 2022), RAP++ (Liu et al., 2021), PrivGSD (Liu et al., 2023), the SOTA method – AIM (McKenna et al., 2022). We refer the reader to the original papers and recent surveys (Yang et al., 2024; Cormode et al., 2025) for details on these methods. Additionally we present the upper bound performance (UB), directly using private dataset without DP guarantees.

**Datasets**. In total, we organize the datasets used in our experiments into four categories. **1) XOR**, simulation stress-test datasets. **2) Structural Causal Model Simulation** generated from causal graphs (details in App. B.1.2). **3) Real-World Datasets with High-Order Correlations**, in which complex classifiers *significantly* outperform simple ones, requiring synthetic data to capture high-order dependencies. **4) Real-World Datasets with Low-Order Correlations**, widely used in the literature, only low-order correlations are sufficient for high downstream accuracy.

**Evaluation Metrics**. Following previous benchmarks (Chen et al., 2025; Tao et al., 2022), we evaluate the methods using Machine Learning (ML) Downstream Efficiency and Fidelity Error. For the fidelity, we calculate the average of total variation distance (TVD) of single and two-way joint distributions between the synthetic and private datasets. Additionally, we perform evaluations in a unified embedding space, derived from an autoencoder trained on the private data with a reconstruction objective. This high-dimensional space enables us to compare the synthetic and real distributions at the representation level rather than just marginals. We calculate Precision and Recall (Sajjadi et al., 2018) which are widely used in image (Gong et al., 2025) and text domains (Wang et al., 2025). We present the details of the metrics in App. B.2.

**Implementation Details**. We provide additional details and hyperparameters of Tab-PE in App. B.3. For the baselines, we follow the original papers and a recent benchmark (Chen et al., 2025) for the hyperparameter settings. We run all methods on three distinct data splits generated by different random seeds and report the average performance values with corresponding standard deviations. When running an $(\epsilon, \delta)$-DP algorithm on a dataset $D_{\mathrm{priv}}$ of size $|D_{\mathrm{priv}}|$, for all methods, we set $\delta = 1/(|D_{\mathrm{priv}}| \cdot \ln |D_{\mathrm{priv}}|)$, which is a common choice in the DP literature (Dwork et al., 2014).

### 4.2 CURSE OF DIMENSIONALITY

In this experiment, we examine the algorithmic capability of methods in modeling high-order correlation. We construct a simulated XOR dataset where all features are drawn from zero-centered uniform distributions. The label is assigned based on the parity of number of positive feature values. In this dataset, the features themselves are mutually independent; the only dependency lies between the features and the label. This setup represents an extreme case where any single feature can flip the label. Consequently, failing to capture the contribution of only a single feature reduces the per-

formance to random guessing (illustrated in Fig. 10& 9, App. B.1.2). The baseline methods are set up with the ideal degree for marginal queries, i.e., $K = \text{num\_features} + 1$.

Fig. 1 presents the AUC score of the classifier trained on the synthetic data generated by the methods at $\epsilon = 1.0$. As the number of features increases, the classification problem itself becomes more challenging leading to the performance drop of the upper bound – using private data ($\epsilon = \infty$). Intuitively, the number of marginal queries grows exponentially with the correlation order. This is challenging to marginal query-based methods for modeling high-order correlations. Consequently, all the baselines fail completely at 5 features, delivering a downstream performance of random guess. In contrast, Tab-PE successfully yields an AUC score of 0.8 for 5 features. This demonstrates **Tab-PE provides broader support for capturing high-order correlations**.

| Dataset | Method | ML Downstream (↑) | | Fidelity (↓) | | Embedding (↑) | |
|---|---|---|---|---|---|---|---|
| | | Accuracy | Macro F1 | 1-TVD | 2-TVD | Precision | Recall |
| Artificial Characters | *UB* | *80.80* $\pm$ *0.44* | *79.87* $\pm$ *0.65* | *0.031* $\pm$ *0.002* | *0.115* $\pm$ *0.003* | *98.09* $\pm$ *0.15* | *98.66* $\pm$ *0.27* |
| | PrivSyn | 13.83 $\pm$ 0.00 | 2.43 $\pm$ 0.00 | 0.054 $\pm$ 0.002 | 0.223 $\pm$ 0.001 | 13.42 $\pm$ 0.32 | 98.05 $\pm$ 0.43 |
| | PrivMRF | 13.63 $\pm$ 0.28 | 4.72 $\pm$ 3.24 | **0.034** $\pm$ **0.004** | 0.206 $\pm$ 0.005 | 13.93 $\pm$ 0.18 | 97.33 $\pm$ 0.59 |
| | GEM | 10.13 $\pm$ 0.86 | 5.62 $\pm$ 0.46 | 0.243 $\pm$ 0.012 | 0.412 $\pm$ 0.011 | 9.55 $\pm$ 0.75 | 94.57 $\pm$ 1.19 |
| | RAP++ | 33.29 $\pm$ 2.14 | 32.17 $\pm$ 2.11 | 0.211 $\pm$ 0.008 | 0.406 $\pm$ 0.014 | 28.45 $\pm$ 4.49 | 3.77 $\pm$ 1.76 |
| | PrivGSD | 40.36 $\pm$ 1.29 | 39.10 $\pm$ 1.38 | 0.168 $\pm$ 0.007 | 0.314 $\pm$ 0.009 | 26.98 $\pm$ 0.36 | **98.40** $\pm$ **0.22** |
| | AIM | 23.24 $\pm$ 1.48 | 20.17 $\pm$ 1.24 | 0.036 $\pm$ 0.005 | **0.177** $\pm$ **0.004** | 18.82 $\pm$ 0.55 | 98.06 $\pm$ 0.21 |
| | Tab-PE | **49.38** $\pm$ **0.46** | **48.09** $\pm$ **0.71** | 0.173 $\pm$ 0.007 | 0.367 $\pm$ 0.010 | **36.57** $\pm$ **1.51** | 89.77 $\pm$ 3.09 |
| Person Activity | *UB* | *78.01* $\pm$ *0.06* | *54.63* $\pm$ *0.36* | *0.009* $\pm$ *0.001* | *0.033* $\pm$ *0.001* | *98.27* $\pm$ *0.13* | *98.30* $\pm$ *0.07* |
| | PrivSyn | 33.05 $\pm$ 0.00 | 4.52 $\pm$ 0.00 | **0.003** $\pm$ **0.000** | 0.195 $\pm$ 0.000 | 41.87 $\pm$ 0.12 | 97.74 $\pm$ 0.12 |
| | PrivMRF | 51.83 $\pm$ 1.28 | 22.42 $\pm$ 1.01 | 0.004 $\pm$ 0.000 | 0.078 $\pm$ 0.001 | 88.85 $\pm$ 0.37 | 98.11 $\pm$ 0.14 |
| | GEM | 31.85 $\pm$ 1.10 | 5.64 $\pm$ 0.79 | 0.218 $\pm$ 0.018 | 0.357 $\pm$ 0.026 | 55.92 $\pm$ 3.42 | 95.20 $\pm$ 1.35 |
| | RAP++ | 52.72 $\pm$ 0.83 | 26.57 $\pm$ 0.82 | 0.190 $\pm$ 0.004 | 0.353 $\pm$ 0.004 | 59.95 $\pm$ 2.49 | 62.36 $\pm$ 2.81 |
| | PrivGSD | 56.47 $\pm$ 0.36 | 29.25 $\pm$ 0.53 | 0.105 $\pm$ 0.005 | 0.201 $\pm$ 0.008 | 80.06 $\pm$ 0.74 | 93.74 $\pm$ 0.58 |
| | AIM | 59.53 $\pm$ 0.47 | 30.79 $\pm$ 0.32 | **0.003** $\pm$ **0.000** | **0.055** $\pm$ **0.000** | 89.97 $\pm$ 0.24 | **98.73** $\pm$ **0.07** |
| | Tab-PE | **63.72** $\pm$ **0.18** | **35.09** $\pm$ **0.19** | 0.036 $\pm$ 0.006 | 0.126 $\pm$ 0.006 | **90.93** $\pm$ **0.88** | 91.57 $\pm$ 0.38 |

Table 1: $\epsilon = 1.0$. The query degree hyperparameter of baselines vary from 2 to 5, the best-performing results of the baselines are reported.

## 4.3 SIMULATED DATASETS BY STRUCTURAL CAUSAL MODELS (SCM)

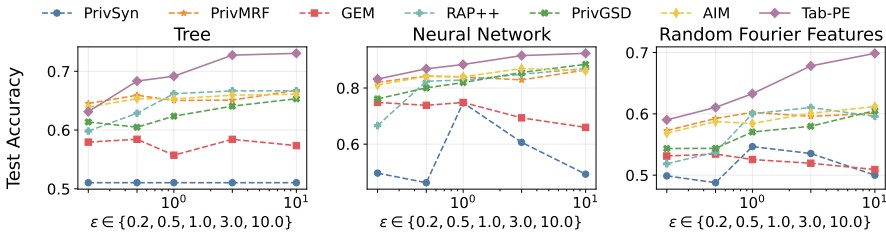

Figure 3: The test accuracy on SCM simulated datasets under various privacy budgets.

We adapt the simulation method from TabPFN (Hollmann et al., 2025), which is a breakthrough in tabular data classification. The full pipeline is described in App. B.1.2. Compared to the previous XOR setting, this is a more realistic scenario: features are correlated; modeling a subset of the joint distribution can translate into gains for downstream tasks. In our experiments, we implement three non-linear prior functions (defining the mapping from features to labels): Tree, Neural Network (NN), and Random Fourier Features (RFF).

Across all prior functions, Tab-PE achieves the best downstream performance at $\epsilon = 1.0$. See Tab. 4, App. C.2 for numerical details. Tab-PE achieves 89.4% accuracy and 96.4% AUC for the neural network prior, significantly above the best baseline – AIM (85.2%, 93.3%). For the fidelity, AIM and MRF offer the best performance, while Tab-PE is slightly behind but still competitive and better than several baselines. In the embedding space, Tab-PE consistently yields the highest

precision ∼98% but the recall slightly lags at ∼81%. Overall, these results indicate that Tab-PE most effectively captures high-order correlations to deliver the highest predictive downstream utility.

Fig. 3 depicts the test accuracy under different settings of privacy budget. In general, Tab-PE consistently outperforms the baselines under a variety of privacy settings. Most methods improve with larger $\epsilon$. Tree and RFF priors induce sharp, brittle high-order correlations that marginal-based methods cannot approximate well. This results in large accuracy gains of Tab-PE, compared to the best-performing baselines, around 10%. In contrast, the NN prior often produces smoother correlations, so the gap remains around 4%. These results demonstrate that Tab-PE is effective at modeling challenging high-order correlations and maintains significant performance gains over baselines under either strict or loose privacy settings.

## 4.4 REAL-WORLD DATASETS

We evaluate on two real-world datasets with high-order correlations (details in App. B.1). Generally, the performance trends are consistent with the previous SCM simulated datasets, as shown in Tab. 1. Tab-PE improves the downstream utilities by a large margin, e.g., +9.02% accuracy and +8.99% macro F1 on the Artificial Characters dataset, but still lags the non-private upper bound (∼30% accuracy gap). Moreover, consistent with the SCM datasets, Tab-PE achieves the highest precision in the embedding space. However, the TVD metrics and recall are slightly worse than AIM and PrivMRF. Moreover, Fig. 4 illustrates the test accuracy under

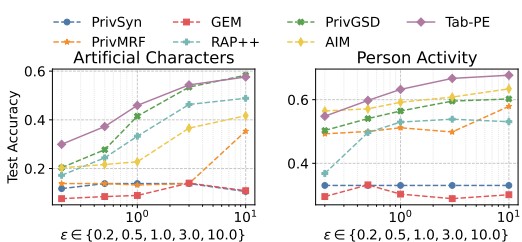

Figure 4: The test accuracy on real-world datasets under various privacy budgets.

different privacy budgets. Tab-PE consistently outperforms the baselines across the privacy settings. Due to space constraints, we present the results of low-order real-world datasets in App. C.3 (Tab. 5). While Tab-PE is primarily designed for high-order correlations, it remains competitive (only ∼1% accuracy drop compared to AIM) on datasets dominated by low-order correlations.

## 4.5 COMPUTE EFFICIENCY & SCALABILITY

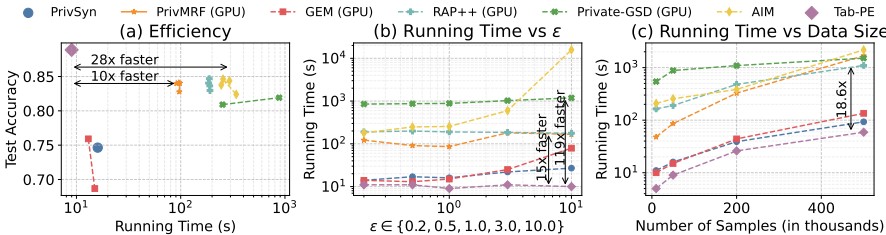

Figure 5: The runtime of the methods under different privacy budgets and dataset sizes. In the left figure, each method is shown with multiple markers, corresponding to various query degree settings. PrivSyn has only one marker as it does not have this hyperparameter.

We further study the compute efficiency and scalability of the methods. We run the experiments on the Neural Network prior simulation dataset using the same computing resources allocated by Slurm: 16 CPUs, 100 GB of memory, and a Quadro RTX 8000 GPU (48GB). While most baselines require GPUs, **Tab-PE runs entirely on CPUs**. As shown in Fig. 5 (left), at $\epsilon = 1.0$, Tab-PE is the most efficient method while achieving the best downstream utilities. Compared to the leading baselines in utility, Tab-PE runs 10× faster than PrivMRF and 28× faster than AIM. We also study the scalability of the baselines by varying the query degree, detailed in App. C.4, Fig. 13. Generally, increasing the query degree does not bring significant performance gains for the baselines. However, it leads to an exponential increase in runtime for GEM and PrivGSD. Subsequently, most methods including Tab-PE scale well with the privacy budget, as presented in Fig. 5 (middle). Meanwhile,

AIM exhibits a rapid increase ($60\times$) in runtime as $\epsilon$ increases from 1.0 to 10.0, as the larger privacy budget allows them to issue more queries. Finally, we examine the scalability of the methods with the dataset size. As depicted in Fig. 5 (right), at $\epsilon = 1.0$, Tab-PE is the fastest method across all dataset sizes. Notably, Tab-PE runs $18.6\times$ faster than the leading baselines (AIM, RAP++, GSD, and MRF) at 500K samples. Taken together, these results demonstrate that **Tab-PE is highly efficient and scalable, demonstrating it is practical for large-scale real-world applications**.

### 4.6 FINDINGS & ANALYSES

**Two-stage selection outperforms ranking- or sampling-only strategies**. Fig. 6 presents the ablation study on two-stage selection by comparing with ranking-only and sampling-only strategies. While the sampling selection can preserve the distribution quickly that translates to TVD-related metrics, the ranking selection is essential for local refinement to boost the downstream accuracy. The two-stage selection effectively combines the advantages of both strategies, leading to the best performance.

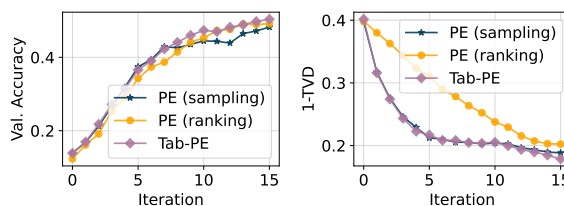

Figure 6: The performance of different selection strategies. Tab-PE implements a two-stage strategy: 5 iterations for sampling and 10 iterations for ranking (Artificial Characters; $\epsilon = 1.0$).

**Polynomial schedule outperforms linear decay**. We study the impact of different probability decay schedules in the `VARIATION_API`. As shown in Fig. 15, App. C.5.1, the polynomial schedule consistently outperforms the linear decay, used in Lin et al. (2025), across all metrics. The polynomial schedule allows more aggressive exploration at the beginning and more focused refinement at the end, leading to better overall performance, illustrated in Fig. 14, App. C.5.1. We present additional performance analysis on different decay factors in App. C.5.1 (Fig. 20 and 21). Generally, a moderate initial mutation rate $\mu_{\text{init}}$ (0.5-0.7) and decay factor $\gamma$ (0.2 - 0.5) yield the best performance and consistently outperform the linear decay ($\gamma = 1.0$).

**Simple APIs can be effective**. We adapt the genetic algorithm design (crossover and mutation) from PrivGSD (Liu et al., 2023) to `VARIATION_API` in our private evolution framework. The results and detailed implementations are provided in App. C.5.2 (Fig. 16). Tab-PE with either API achieves higher accuracy compared to the best marginal-based method ($\sim$40%). The simple random walk with scheduled probability decay boosts accuracy by 7%, from 43% to 50%, compared to crossover and mutation, while the TVD metrics remain competitive.

**Hyperparameter Sensitivity Analysis**. We study the sensitivity of key hyperparameters in Tab-PE. The detailed results are presented in App. C.6. Generally, Tab-PE needs sufficient iterations (15-20) to converge and provide good utilities. Our ideal number of iterations is notably smaller than PrivGSD, which performs 200K iterations. The number of synthetic samples should be proportional to the dataset size to ensure an appropriate signal-to-noise ratio. At $\epsilon = 1.0$, Tab-PE best generates 10-20% of the original size (Fig. 17), but the synthetic data can be further enriched by oversampling algorithms (App. C.7). The optimal hyperparameter setting is robust across $\epsilon$ settings (Fig. 23).

## 5 CONCLUSION

We revisit the challenges of modeling high-order correlations in synthetic tabular data generation with DP guarantees. We showed that existing methods struggle to capture these correlations. To address this, we introduced Tab-PE, a novel approach using Private Evolution. Our method effectively models high-order correlations while being lightweight and efficient. While Private Evolution has enhanced the performance in image and text synthesis (Lin et al., 2024; Xie et al., 2024; Lin et al., 2025), a prior attempt for tabular data (Swanberg et al., 2025) did not yield satisfactory outcomes. In contrast, our results demonstrate that with appropriate design choices, Private Evolution can offer some advantages in either utility or efficiency over existing methods. We believe our work establishes a new promising paradigm for private tabular data generation.

## REPRODUCIBILITY STATEMENT

Our experimental details are fully described in the main paper and Appendix. The code, datasets, and instructions are available at `https://anonymous.4open.science/r/tabpe-A11C`

## ETHICAL STATEMENT

We believe our work has positive ethical implications. By enabling the generation of high-quality synthetic tabular data with differential privacy guarantees, our methods can enable data sharing, application, and innovation in many fields where privacy concerns currently limit data access. However, we also acknowledge the potential risks of synthetic data, even with DP guarantees, loose settings of privacy parameters may still lead to information leakage. We encourage users to carefully consider the privacy-utility trade-offs and choose appropriate privacy parameters for their specific use cases.

Large Language Models were occasionally used for polishing, the vast majority of the writing was done manually.

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

# Appendix

Due to the space limit, we present additional details, results, and analyses in the Appendix.

# A METHODOLOGY

## A.1 PRIVACY ANALYSIS

The privacy analysis of Algorithm 2 can be reused from Lin et al. (2024) (see Section 4.3 there), as Tab-PE changes only non-private steps of the original PE framework. For completeness, we include it here as well. The DP guarantee of the Tab-PE algorithm (Algorithm 2) can be reasoned as follows:

- Step 1: The sensitivity of DP Nearest Neighbors Histogram (Algorithm 1). Each private sample only contributes one vote for histogram of one class. If we add or remove one sample, the resulting histogram for the corresponding class will change by at most 1 in the $\ell_2$ norm. Therefore, the sensitivity is upper bounded by 1.

- Step 2: Regarding each PE iteration as a Gaussian mechanism. The second for loop of Algorithm 1 adds i.i.d. Gaussian noise with standard deviation $\sigma$ to each bin. This is a standard Gaussian mechanism (Dwork et al. (2014)) with noise multiplier $\sigma$.

- Step 3: Regarding the entire PE algorithm as $T$ adaptive compositions of Gaussian mechanisms, as Tab-PE is simply applying Algorithm 1 $T$ times sequentially.

- Step 4: Regarding the entire Tab-PE algorithm as one Gaussian mechanism with noise multiplier $\sigma/\sqrt{T}$. It is a standard result from Dong et al. (2022) (see Corollary 3.3 therein).

- Step 5: Computing DP parameters $\epsilon$ and $\delta$. Since the problem is simply computing $\epsilon$ and $\delta$ for a standard Gaussian mechanism, we use the formula from Balle & Wang (2018) directly.

# B EXPERIMENTAL SETUP

## B.1 DATASETS

### B.1.1 REAL-WORLD DATASETS

**Qualifying high-order correlation through classifier performance gap**  We aim to study how well the methods can capture high-order correlations. It is easy to be misled about high-dimensional correlations and high-dimensional datasets. While some datasets can have a large number of features, but the features are often independent or only have low-order correlations (i.e., dependencies involving only a few features). We first propose a way to qualify the order of correlation in a dataset by considering the performance gap between simple classifiers, which only capture low-order correlations, and complex classifiers, which can leverage high-order correlations. The larger gap, the more high-order correlations exist in the dataset. In practice, we vary the max depth of XGBoost, where the depth of decisions work as an upper bound on the order of captured correlations.

**Widely used datasets are dominated by low-order correlations**  We investigate a variety of datasets that have been widely used in prior evaluations (Chen et al., 2025; Tao et al., 2022). We increase the max depth from 2 to 7, while keeping other hyperparameters as default. The results are shown in Figure 7. The gap of accuracy is trivial (typically smaller than 1%). This indicates that the downstream tasks on these datasets are dominated by low-order correlations. This leads to the conclusion that these datasets are not suitable for evaluating the ability to capture high-order correlations because synthesizers that can only capture low-order correlations may already achieve good performance.

**Datasets with high-order correlations**  We selected two datasets that yield significant differences in accuracy while varying the max depth of XGBoost, as depicted in Figure 8. In particular, we consider the Artificial Characters dataset (Guvenir et al., 1992) [3] and the Person Activity dataset (Vidulin et al., 2010) [4]. The Artifical Characters dataset contains 10218 samples with 8 numerical features and 10 classes, while the Person Activity dataset includes 164860 samples with 2 categorical features, 6 numerical features, and 11 classes.

---

[3] https://www.openml.org/search?type=data&id=1459
[4] https://www.openml.org/search?type=data&id=1483

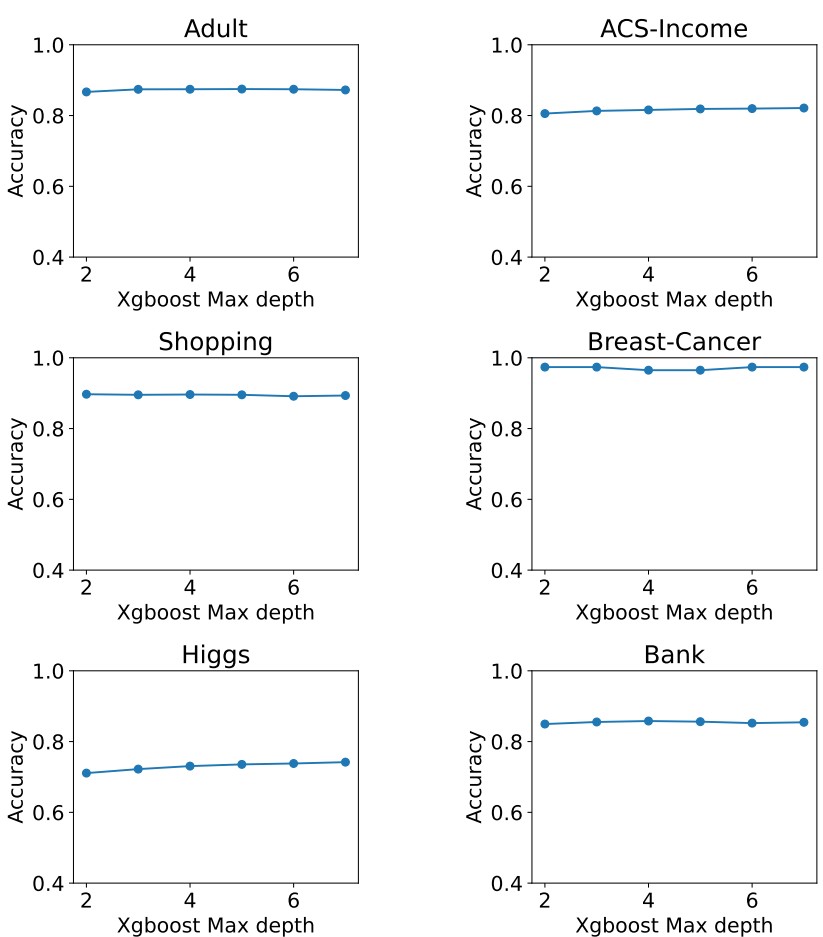

Figure 7: Datasets with low-order correlations. These are widely used in prior evaluations.

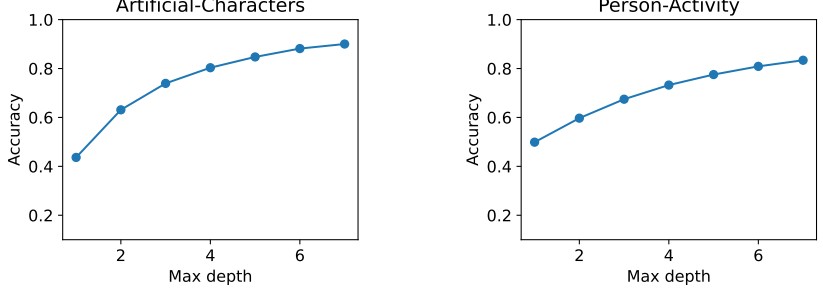

Figure 8: Datasets with high-order correlations – our focus.

### B.1.2  SIMULATION DATASETS

**XOR correlations as a stress test**  We consider XOR correlations as a stress test for capturing high-order correlations. The XOR function is a classic example that requires all input features to determine the output. Failing to model any single feature leads to random guessing. Each feature is drawn from an uniform distribution over $(-10, 10)$. The label is then determined by the parity of positive features.

$$c = \begin{cases} 1 & \text{if } \sum_{i=1}^{d} \mathbb{1}(x_i > 0) \text{ is odd} \\ 0 & \text{otherwise} \end{cases}$$

For each setting of the number of features, we generate 50K samples and ensure balanced binary classes. The dataset with two features is visualized in Figure 9. Figure 10 presents the performance of XGBoost classifiers with varying max depths on the XOR datasets. The max depth of XGBoost must be equal to the number of features to achieve better-than-random accuracy. Therefore, the synthetic data must capture the full high-order correlations to achieve good downstream utilities.

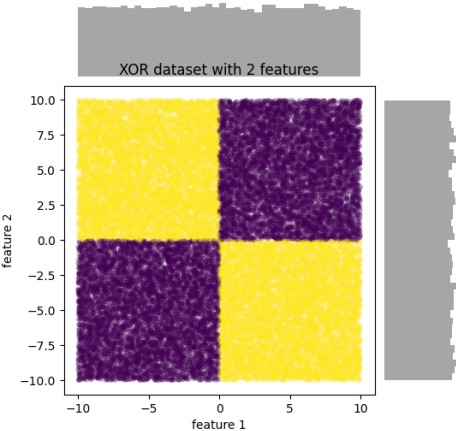

Figure 9: XOR dataset with 2 features. The colors represent classes.

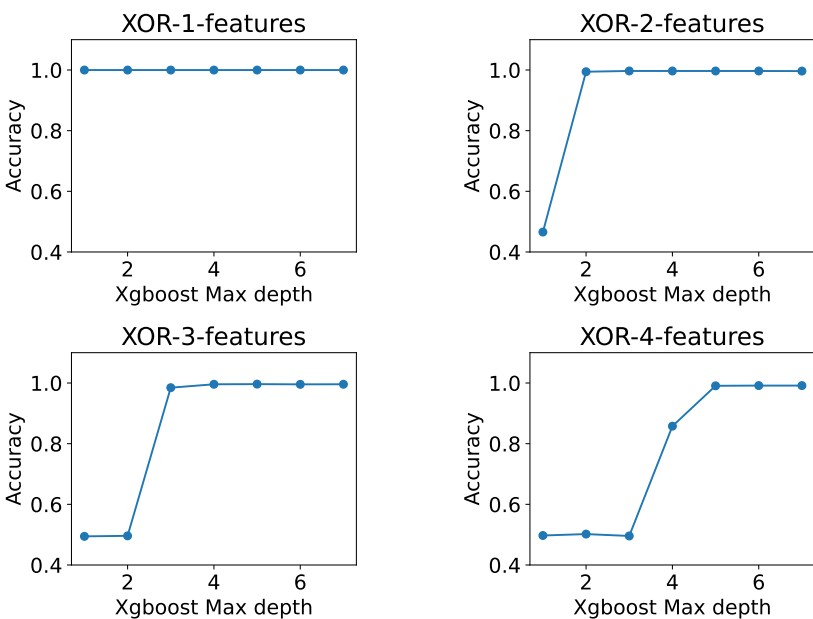

Figure 10: XOR Simulation Datasets.

**SCM simulation datasets offer sustainable high-order correlations**   We adapt the simulation method from TabPFN (Hollmann et al., 2025), which is a breakthrough in tabular data classification. TabPFN generates large-scale realistic simulation data and pretrains a foundation model for in-context learning. By learning on only the simulation data, TabPFN still offers strong generalization to real-world data. This simulation pipeline employs Structural Causal Models (SCMs). An SCM defines a directed acyclic graph where each node corresponds to a feature, and the edges capture causal dependencies. The features are then generated by sampling values according to these dependencies that can represent complex interactions and non-linear relationships. The label is calculated by a prior function of features, inducing high-order correlations between the label and the feature set. As a result, increasing the max depth of XGBoost can lead up to a 10% accuracy gap (Figure 11, Appendix B.1). Compared to the previous XOR setting, this is a more realistic scenario: features are correlated; modeling a subset of the joint distribution can translate into gains for downstream tasks. In our experiments, we implement three non-linear prior function: Tree, Neural Network (NN), and Random Fourier Features (RFF). Each dataset include 50K samples.

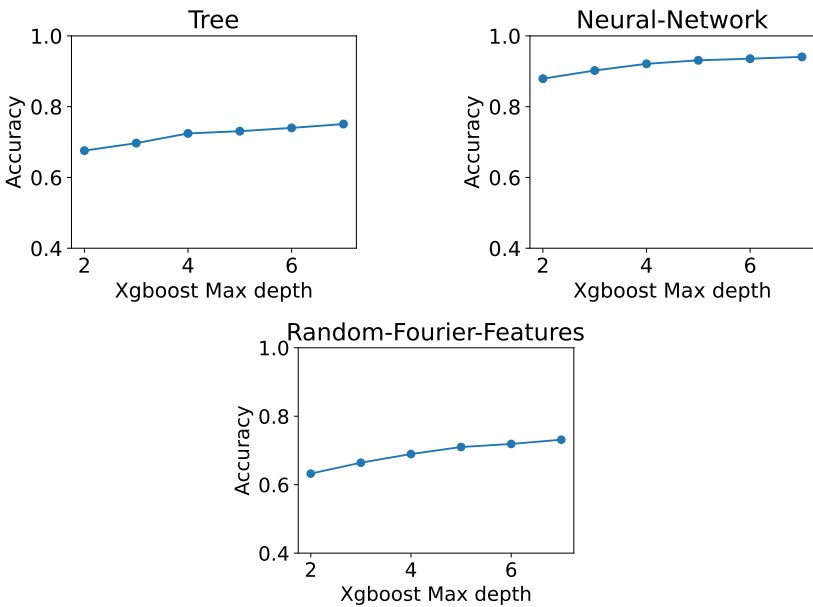

Figure 11: SCM Simulation Datasets.

B.2   EVALUATION METRICS

We consider the following metrics to evaluate the quality of synthetic data.

**Downstream utility**   The downstream utility reflects how well the synthetic data capture the correlation between features and labels. These metrics are the most important ones for studying high-order correlations. For consistency, we use the same SOTA classifier TabICL (Qu et al., 2025) for all datasets. TabICL is a transformer-based foundation model that has been pretrained on 82 millions of tabular datasets with in-context learning. It has demonstrated superior performance on a wide range of tabular datasets while **requiring little-to-no hyperparameter tuning**. For all methods, we fit TabICL on the generated synthetic data and evaluate on the *same* real test set.

**Fidelity of statistical properties**   To capture the statistical properties, we consider the total variation distance (TVD) of the pairwise joint distributions, as used in prior work (Chen et al., 2025; Tao et al., 2022). In particular, we bin each numerical feature into 20 equal-width bins. We consider 1-TVD and 2-TVD, which are the average TVD of all univariate and bivariate distributions, respectively. The following formula defines the TVD metrics:

$$1\text{-TVD}(\mathcal{D}, \mathcal{D}') = \frac{1}{2N_{\mathcal{F}}} \sum_{f_i \in \mathcal{F}} \sum_{v \in f_i} |P_{\mathcal{D}}(x_i = v) - P_{\mathcal{D}'}(x_i = v)|$$

$$2\text{-TVD}(\mathcal{D}, \mathcal{D}') = \frac{1}{2\,N_{\mathcal{F}}(N_{\mathcal{F}} - 1)} \sum_{\substack{f_i, f_j \in \mathcal{F} \\ i \neq j}} \sum_{v_i \in f_i} \sum_{v_j \in f_j}$$

$$\times \left| P_{\mathcal{D}}(x_i = v_i, x_j = v_j) - P_{\mathcal{D}'}(x_i = v_i, x_j = v_j) \right|$$

where $\mathcal{F}$ is the set of features/attributes including the label, $N_{\mathcal{F}}$ is the number of features, and $P_{\mathcal{D}}$ and $P_{\mathcal{D}'}$ are the empirical probability by counting within datasets $\mathcal{D}$ and $\mathcal{D}'$, respectively.

**Representation-level alignment**   Evaluating the alignment on the representation space is common for text and image generation Lin et al. (2024); Xie et al. (2024). The alignment reflects how well the synthetic data cover the real data distribution in the representation space which can capture somewhat high-dimensional dependencies. While the representation space is achieved directly from foundation models in text and image domains, tabular data is challenged by strong distribution-shift across datasets. Therefore, we train an autoencoder for each dataset using directly the private dataset with a reconstruction loss. It is worth noting that the autoencoder here is only used for evaluation, and is not part of the synthesis process. By training on the private data, we ensure that the representation space is reliable and meaningful. We then calculate precision and recall Sajjadi et al. (2018) on the embeddings of real and synthetic data. Precision measures how many generated samples are actually close to the real data manifold, while Recall calculates how many real samples are covered by the generated data. The formula of precision and recall are as follows:

$$\text{Precision} = \frac{1}{|\mathcal{D}_{\text{syn}}|} \sum_{x \in \mathcal{D}_{\text{syn}}} \mathbb{1}\left(\exists y \in \mathcal{D}_{\text{real}}, \|\phi(x) - \phi(y)\|_2 \leq r_k(\phi(y), \phi(\mathcal{D}_{\text{real}}))\right)$$

where $\phi$ is the encoder of the autoencoder that maps the raw data to the representation space; $r_k(\phi(y), \phi(\mathcal{D}_{\text{real}}))$ is the distance from $\phi(y)$ to its $k$-th nearest neighbor in the set $\phi(\mathcal{D}_{\text{real}})$. We set $k = 5$ in our experiments. Recall is defined symmetrically by swapping $\mathcal{D}_{\text{syn}}$ and $\mathcal{D}_{\text{real}}$.

$$\text{Recall} = \frac{1}{|\mathcal{D}_{\text{real}}|} \sum_{y \in \mathcal{D}_{\text{real}}} \mathbb{1}\left(\exists x \in \mathcal{D}_{\text{syn}}, \|\phi(y) - \phi(x)\|_2 \leq r_k(\phi(x), \phi(\mathcal{D}_{\text{syn}}))\right)$$

### B.3    IMPLEMENTATIONS

We split each dataset into 70% training, 15% validation, and 15% test sets, determined by fixed random seeds. All the methods are fitted on the same training set and evaluated on the same test set. The validation set is used for hyperparameter tuning for all methods. We generally do not account the privacy budget for hyperparameter tuning. For the baselines, we reuse the code from a recent benchmark (Chen et al., 2025) and follow their hyperparameter settings. For baselines that requires discretizing numerical features, we employ PrivTree (Zhang et al., 2016), which yields better performance than uniform binning, according to Chen et al. (2025). For baselines using statistical queries, we use marginal queries, as they are the most commonly used in prior work and the most important for capturing high-order correlations. Rather than fixing the degree of marginal queries at 2, as is common in many previous setups, we treat it as a tunable hyperparameter (ranging from 2 to 5), since our datasets exhibit high-order correlations. This tuning maximizes the chance of capturing such correlations. The other hyperparameters of the baselines are presented in Table 2.

By default, we run Tab-PE with the hyperparameters presented in Table 3 if not specified. For the real-world datasets, we generate 1K samples for the Artifical Characters dataset and 5K samples for the Person Activity dataset. For the simulation data, we generate 2K samples by default.

## C    ADDITIONAL RESULTS

### C.1    DATA DISTRIBUTION OF TAB-PE OVER ITERATIONS

Figure 12 illustrates the evolutionary process of synthetic datasets generated by Tab-PE. At the beginning (iteration 0), the synthetic data is mostly random. As the algorithm progresses, the synthetic data gradually aligns with the private data distribution.

| Method | Hyperparameter | Value |
|---|---|---|
| PrivSyn | Consistent Iteration | 501 |
| | Max update iteration | 50 |
| PrivMRF | Graph construction parameter | 6 |
| | Sample size | 400 |
| | Estimation iteration | 3000 |
| | Size penalty | 1e-8 |
| | Max clique size | 1e+7 |
| GEM | Synthesis size | 1024 |
| | Learning rate | 1e-3 |
| | Max iteration | 500 |
| | Max selection round | $5 \cdot$ number of attributes |
| RAP++ | Random Projection Number | 2e+6 |
| | Categorical optimization rate | 3e-3 |
| | Numerical optimization rate | 6e-3 |
| | Top q | 5 |
| | Categorical optimization step | 1 |
| | Numerical optimization step | 3 |
| | Upsample rate | 10 |
| PrivGSD | Mutation rate | 50 |
| | Cross over rate | 50 |
| | Upsample number | 1e+5 |
| | Number of iterations | 1e+6 |
| AIM | Max model size | 100 |
| | Max iteration | 1000 |
| | Max marginal size | 2.5e+5 |

Table 2: Hyperparameters of the baselines.

| Hyperparameter | Value |
|---|---|
| Number of iterations $T$ | 15 |
| Number of sampling iterations $T_{\text{sampling}}$ | 5 |
| Variation degree $m$ | 3 |
| Mutation rate initial value $\mu_{\text{init}}$ | 0.5 |
| Mutation rate final value $\mu_{\text{final}}$ | 0.02 |
| Categorical mutation rate $\mu_{\text{cat}}$ | Polynomial decay from $\mu_{\text{init}}$ to 0.02 |
| Numerical mutation rate $\mu_{\text{num}}$ | Polynomial decay from $\mu_{\text{init}}$ to 0.02 |
| Decay factor $\gamma$ | 0.2 |
| Categorical weight $\lambda$ | 1/3 |
| Privacy budget $\epsilon$ | 1.0 |
| Privacy delta $\delta$ | $1/(|\mathcal{D}_{\text{real}}| \cdot \ln(|\mathcal{D}_{\text{real}}|))$ |

Table 3: Default hyperparameters of Tab-PE.

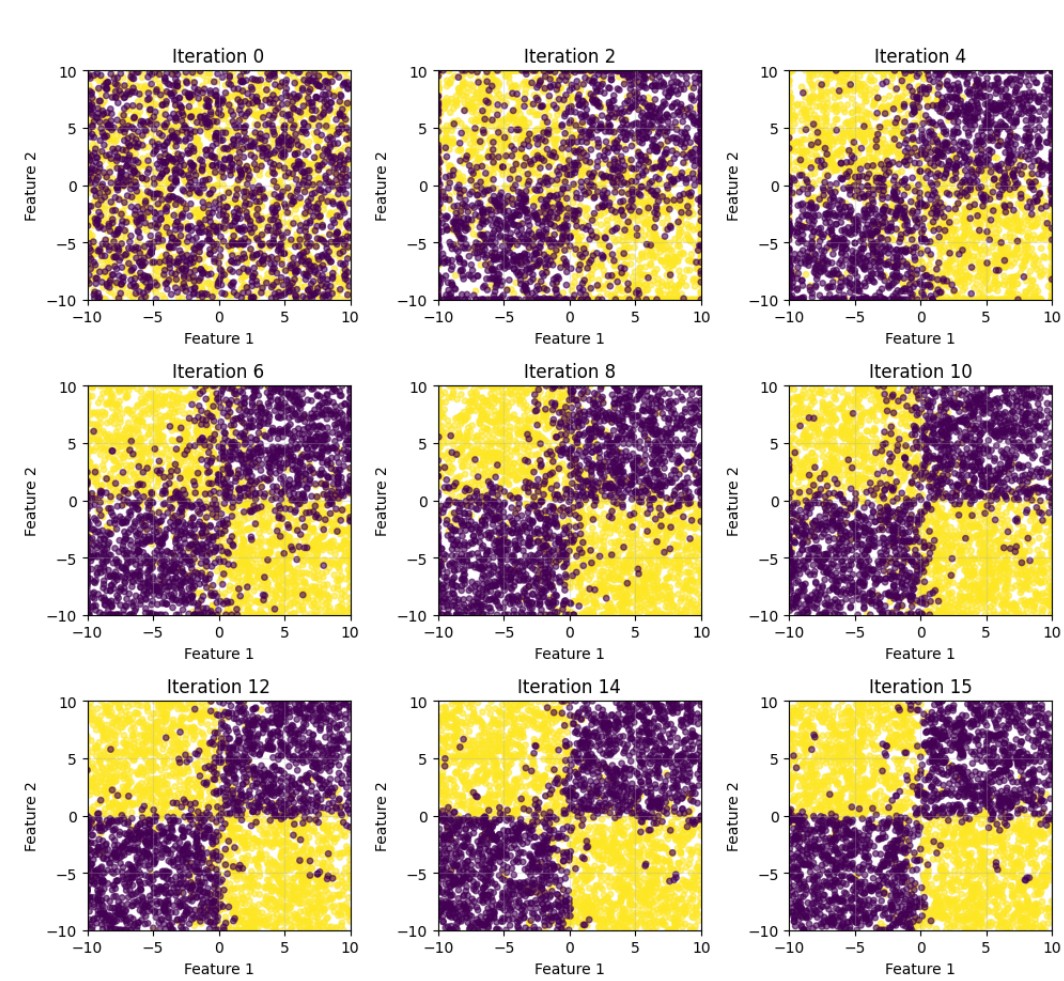

Figure 12: Synthetic Datasets generated by Tab-PE over iterations for the XOR dataset with 2 features.

| Prior | Method | ML Downstream (↑) | | Fidelity (↓) | | Embedding (↑) | |
|---|---|---|---|---|---|---|---|
| | | Accuracy | AUC Score | 1-TVD | 2-TVD | Precision | Recall |
| Tree | *UB* | *81.41* ±0.09 | *90.17* ±0.45 | *0.017* ±0.000 | *0.062* ±0.001 | *98.05* ±0.22 | *97.74* ±0.09 |
| | PrivSyn | 51.04 ±0.00 | - | 0.037 ±0.001 | 0.129 ±0.002 | 65.61 ±0.98 | 98.08 ±0.17 |
| | PrivMRF | 65.68 ±0.52 | 71.26 ±0.82 | **0.037** ±0.002 | 0.085 ±0.002 | 92.11 ±0.61 | **98.34** ±0.13 |
| | GEM | 56.66 ±0.67 | 60.47 ±0.47 | 0.119 ±0.009 | 0.224 ±0.013 | 59.65 ±1.15 | 97.92 ±0.18 |
| | RAP++ | 64.77 ±1.09 | 69.77 ±1.49 | 0.152 ±0.003 | 0.254 ±0.006 | 93.21 ±1.11 | 23.94 ±3.32 |
| | PrivGSD | 61.99 ±0.37 | 66.22 ±0.85 | 0.126 ±0.003 | 0.208 ±0.004 | 84.79 ±0.77 | 91.74 ±0.28 |
| | AIM | 65.40 ±0.26 | 71.19 ±0.10 | **0.037** ±0.001 | **0.083** ±0.002 | 93.47 ±0.45 | 98.33 ±0.04 |
| | Tab-PE | **68.78** ±0.30 | **75.24** ±0.36 | 0.064 ±0.002 | 0.165 ±0.002 | **98.63** ±0.31 | 79.61 ±1.33 |
| NN | *UB* | *96.58* ±0.10 | *96.58* ±0.10 | *0.016* ±0.000 | *0.062* ±0.000 | *98.02* ±0.16 | *97.77* ±0.05 |
| | PrivSyn | 51.97 ±16.18 | 50.59 ±22.57 | **0.039** ±0.003 | 0.136 ±0.007 | 66.31 ±0.08 | 97.93 ±0.36 |
| | PrivMRF | 84.78 ±0.71 | 92.75 ±0.77 | **0.039** ±0.004 | 0.087 ±0.006 | 92.89 ±0.29 | **98.41** ±0.25 |
| | GEM | 74.26 ±0.81 | 82.61 ±0.47 | 0.136 ±0.007 | 0.245 ±0.011 | 57.55 ±0.76 | 98.24 ±0.13 |
| | RAP++ | 85.16 ±1.21 | 93.06 ±1.10 | 0.152 ±0.002 | 0.256 ±0.002 | 94.00 ±0.26 | 21.81 ±2.22 |
| | PrivGSD | 82.47 ±0.40 | 90.86 ±0.49 | 0.129 ±0.005 | 0.214 ±0.007 | 84.96 ±0.42 | 91.17 ±0.56 |
| | AIM | 85.23 ±0.40 | 93.26 ±0.57 | 0.058 ±0.002 | 0.159 ±0.001 | 93.10 ±0.26 | 98.36 ±0.23 |
| | Tab-PE | **89.36** ±0.42 | **96.37** ±0.25 | 0.048 ±0.000 | 0.137 ±0.001 | **98.57** ±0.39 | 81.27 ±1.75 |
| RFF | *UB* | *81.12* ±0.19 | *81.12* ±0.19 | *0.017* ±0.000 | *0.063* ±0.001 | *98.12* ±0.18 | *97.55* ±0.16 |
| | PrivSyn | 50.96 ±2.61 | 50.56 ±4.13 | **0.035** ±0.002 | 0.122 ±0.004 | 66.83 ±0.59 | 97.77 ±0.44 |
| | PrivMRF | 60.11 ±0.15 | 63.68 ±0.29 | 0.037 ±0.002 | 0.083 ±0.003 | 93.06 ±0.16 | **98.55** ±0.07 |
| | GEM | 53.76 ±0.99 | 55.53 ±0.95 | 0.133 ±0.022 | 0.243 ±0.032 | 57.80 ±3.26 | 98.51 ±0.21 |
| | RAP++ | 59.00 ±1.32 | 62.13 ±1.72 | 0.001 ±0.000 | 0.162 ±0.004 | 0.270 ±0.007 | 25.84 ±2.99 |
| | PrivGSD | 57.08 ±0.07 | 59.70 ±0.15 | 0.135 ±0.002 | 0.225 ±0.005 | 85.13 ±0.24 | 90.80 ±0.79 |
| | AIM | 59.60 ±1.32 | 62.76 ±1.41 | **0.035** ±0.002 | **0.079** ±0.003 | 93.41 ±0.31 | 98.50 ±0.16 |
| | Tab-PE | **64.10** ±0.40 | **69.19** ±0.45 | 0.058 ±0.002 | 0.159 ±0.001 | **98.57** ±0.39 | 81.27 ±1.75 |

Table 4: The experiment is configured with $\epsilon = 1.0$. The degree hyperparameter of baselines varies from 2 to 5. The best and second-best results are highlighted in **bold** and underline, respectively.

## C.2 SCM Simulation Datasets

Table 4 presents the performance of all methods on the SCM simulation datasets. Tab-PE consistently outperforms all baselines for the downstream utility metrics. For the fidelity metrics, Tab-PE offers competitive perfomance. AIM remains the best on 1-TVD and 2-TVD, as it is designed for capturing low-order marginals. In the embedding space, Tab-PE achieves the best precision, demonstrating that the synthetic samples generated by Tab-PE are indeed close to the real data at the representation level. Overall, these results indicate the effectiveness of Tab-PE in capturing high-order correlations.

## C.3 Real-world Datasets with Low-Order Correlations

We further evaluate the methods on well-known real-world datasets with low-order correlations. Compared to Adult, Breast Cancer is a more challenging dataset with 30 features and only ∼500 samples. In this setting, we configure Tab-PE to run for 30 iterations generating 2K samples and 20 iterations generating 100 samples, respectively for the Adult and Breast Cancer datasets. The results are presented in Table 5. Consistent to the prior evaluations (Chen et al., 2025), AIM offers the leading performance across most metrics. Tab-PE remains competitive on the downstream utilities with only 1% accuracy drop compared to AIM. For low-order correlations, the marginal-based methods are sufficient to capture the essential relationships between features and labels. This explains why AIM, RAP, GSD, and PrivMRF perform well on these datasets. Overall, these results indicate that while Tab-PE is primarily designed for high-order correlations, it remains competitive on datasets dominated by low-order correlations.

## C.4 Compute Efficiency

We present the running time and test accuracy of the baselines while varying the degree of marginal queries in Figure 13. Generally, increasing the degree of marginal queries does not significantly improve the accuracy. As the degree increases, the number of queries grows exponentially. For PrivMRF, the test performance peaks at the degree of 4 at 84% and remains stable at 83.5%. For

| Dataset | Method | ML Downstream (↑) | | Fidelity (↓) | | Embedding (↑) | |
|---|---|---|---|---|---|---|---|
| | | Accuracy | Macro F1 | 1-TVD | 2-TVD | Precision | Recall |
| Adult | UB | 84.41 ± 0.57 | 75.68 ± 1.54 | 0.010 ± 0.001 | 0.027 ± 0.001 | 94.50 ± 0.14 | 94.09 ± 0.29 |
| | PrivSyn | 75.77 ± 0.00 | 43.11 ± 0.00 | 0.012 ± 0.001 | 0.086 ± 0.000 | 45.34 ± 1.54 | 89.34 ± 0.17 |
| | PrivMRF | 83.15 ± 0.43 | **76.85** ± **0.64** | 0.008 ± 0.000 | 0.034 ± 0.000 | 84.15 ± 0.25 | **93.74** ± **0.21** |
| | GEM | 79.17 ± 2.32 | 69.63 ± 1.48 | 0.009 ± 0.006 | 0.086 ± 0.002 | 0.185 ± 0.004 | 76.48 ± 2.32 |
| | RAP++ | 80.87 ± 0.59 | 72.22 ± 1.25 | 0.063 ± 0.000 | 0.137 ± 0.002 | 61.08 ± 4.24 | 80.64 ± 1.53 |
| | PrivGSD | 82.09 ± 0.11 | 75.90 ± 0.43 | 0.028 ± 0.001 | 0.069 ± 0.001 | 74.45 ± 0.47 | 80.81 ± 0.41 |
| | AIM | **83.36** ± **0.41** | 76.10 ± 1.41 | **0.007** ± **0.001** | **0.032** ± **0.001** | 87.06 ± 0.75 | 93.18 ± 0.21 |
| | Tab-PE | 82.22 ± 0.51 | 73.66 ± 0.87 | 0.120 ± 0.008 | 0.221 ± 0.017 | 34.27 ± 1.57 | 77.25 ± 7.42 |
| Breast Cancer | UB | 97.68 ± 1.64 | 97.56 ± 1.73 | 0.143 ± 0.008 | 0.390 ± 0.015 | 97.48 ± 0.73 | 95.64 ± 0.78 |
| | PrivSyn | 51.60 ± 8.03 | 38.99 ± 6.92 | 0.431 ± 0.014 | 0.704 ± 0.012 | 60.39 ± 7.34 | 21.78 ± 7.84 |
| | PrivMRF | 60.41 ± 3.37 | 37.63 ± 1.33 | 0.394 ± 0.020 | 0.683 ± 0.022 | 51.01 ± 11.29 | 16.83 ± 8.23 |
| | GEM | 50.74 ± 13.88 | 44.91 ± 12.27 | 0.396 ± 0.012 | 0.681 ± 0.010 | **69.60** ± **11.83** | **28.64** ± **8.97** |
| | RAP++ | 84.81 ± 4.43 | 83.97 ± 4.45 | 0.425 ± 0.018 | 0.699 ± 0.021 | 64.62 ± 3.48 | 9.63 ± 4.71 |
| | PrivGSD | 60.02 ± 3.13 | 37.49 ± 1.24 | 0.419 ± 0.017 | 0.686 ± 0.023 | 60.30 ± 5.38 | 21.27 ± 1.91 |
| | AIM | **89.25** ± **4.75** | **87.82** ± **6.17** | 0.419 ± 0.016 | 0.707 ± 0.017 | 68.17 ± 6.03 | 12.65 ± 4.86 |
| | Tab-PE | 88.48 ± 3.53 | 87.01 ± 4.74 | 0.431 ± 0.014 | 0.779 ± 0.015 | 69.02 ± 6.14 | 15.24 ± 3.11 |

Table 5: Comparison on low-order real-world datasets under $\epsilon = 1$. The best and second-best results are highlighted in **bold** and underline, respectively. UB refers to the upper bound performance trained on real data.

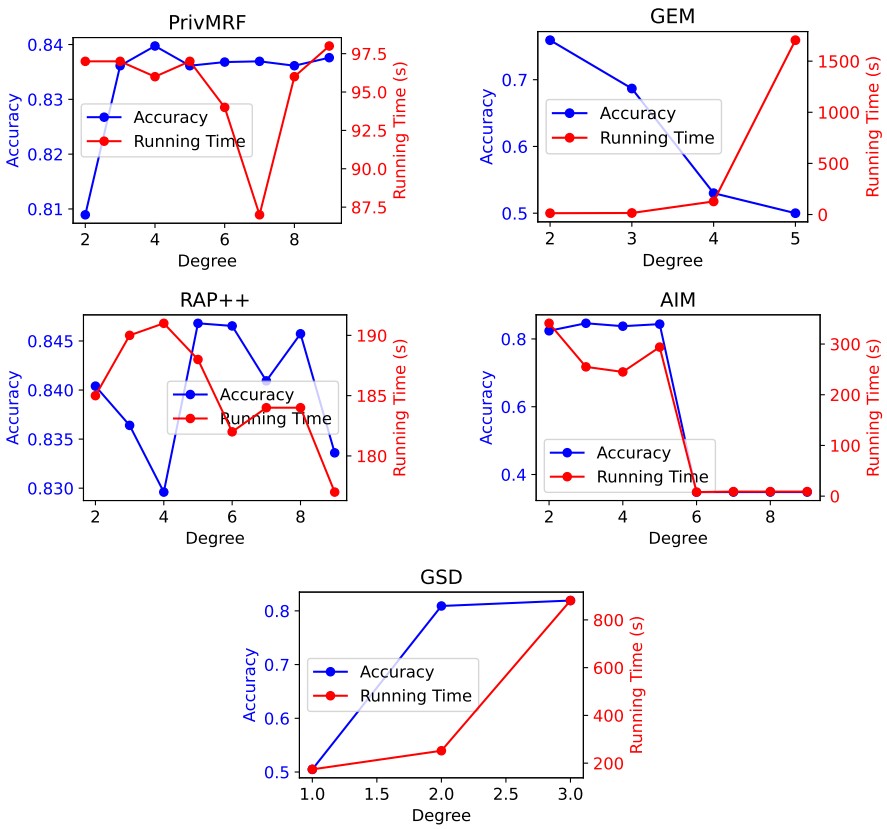

Figure 13: Running Time and Test Accuracy of the baselines while varying the degree of marginal queries. The trivial accuracy (random guessing) is 50%.

GEM, the accuracy drops as the degree increases, due to the high noise to answer the large number of queries. The running time of GEM significantly increases at the degree of 5. For RAP++, the accuracy slightly increases from 84 to 84.45 at the degree of 4 then drops as the degree increases. The running time of RAP++ is not significantly affected by the degree of queries. For AIM, the accuracy remains approximately at 82% for all degrees that are less than 6. A degree that is too high ($\geq 6$) causes the method to collapse without producing any meaningful patterns. For GSD, the maximum degree is 3 due to the high compute resource requirement. In particular, at the degree of 4, GSD requires more than 200GB GPU memory which is not affordable for us. Theoretically, the running time of methods that rely on marginal queries grows exponentially as the number of queries increases. However, in practice, the running time of some methods may be still affordable and do not change significantly. This is because some implementations limit the number of queries to a fixed number to make sure the noise is not too large to yield reliable query answers. As a result, they may not be able to fully model the high-order correlations. In summary, these results indicate the limitations of marginal-based methods of both efficiency and effectiveness in capturing high-order correlations.

### C.5 VARIATION_API STUDY

For consistency, this section presents the results on the Artificial Characters dataset with $\epsilon = 1.0$.

#### C.5.1 DECAY SCHEDULE STUDY

Figure 14 illustrates the linear and polynomial decay schedules for the mutation rate. Generally, the polynomial decay provides a higher mutation rate at the early stage for exploration while maintaining a smaller mutation rate at the later iterations for better refinement. This leads to better performance of the polynomial schedule over the linear decay, as shown in Figure 15.

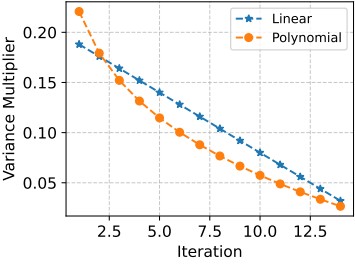

Figure 14: Visualization of different decay schedules.

#### C.5.2 API OPERATORS

We compare the proposed random walk strategy and a genetic algorithm-based design from an existing work – PrivGSD (Liu et al., 2023). It is worth noting there are significant differences between Private Evolution and PrivGSD. PrivGSD is a method that heavily relies on marginal queries. PrivGSD first defines a set of marginal queries and privately answers them. Then it uses a genetic algorithm to search for a synthetic dataset that minimizes the error compared to the noisy answers. Therefore, PrivGSD still inherits the limitations of marginal query-based approaches in capturing high-order correlations. In contrast, Tab-PE does not rely on marginal queries, our evolutionary process is directly guided by the private data at each iteration. In this comparison, we adapt the genetic algorithm-based design from PrivGSD to our VARIATION_API interface. More specifically, the original operators in PrivGSD work at dataset levels, while Tab-PE's VARIATION_API requires sample-level operators. To achieve this, we remove the random selection of samples from the dataset, and instead we apply the operators to all samples in the synthetic dataset. Additionally, PrivGSD performs mutation only one attribute at a time, which leads to very slow convergence ( 200K iterations). This is not affordable for Private Evolution, as the privacy budget is consumed at each iteration. Therefore, we modify the mutation operator to allow all attributes at once. The crossover operator is kept the same as in PrivGSD. Figure 16 presents the results. This confirms that the simple random-walk design in Tab-PE is effective and efficient.

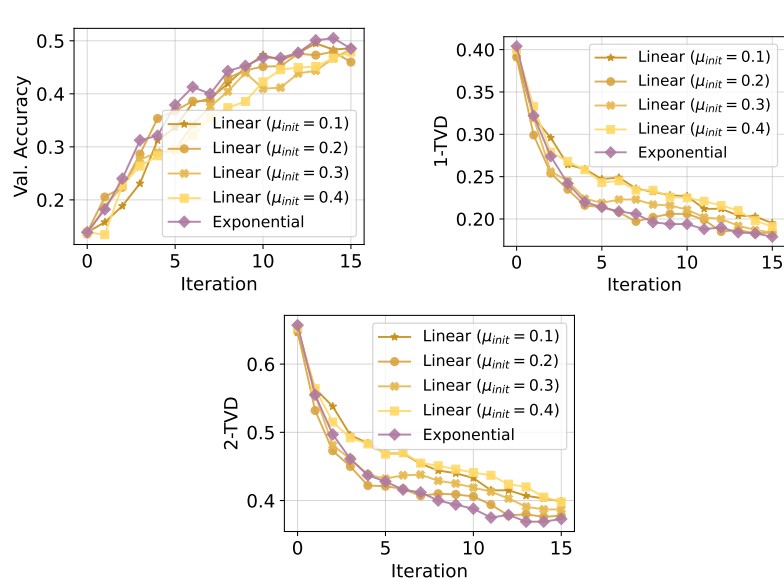

Figure 15: The performance of Tab-PE with different mutation rate decay schedules.

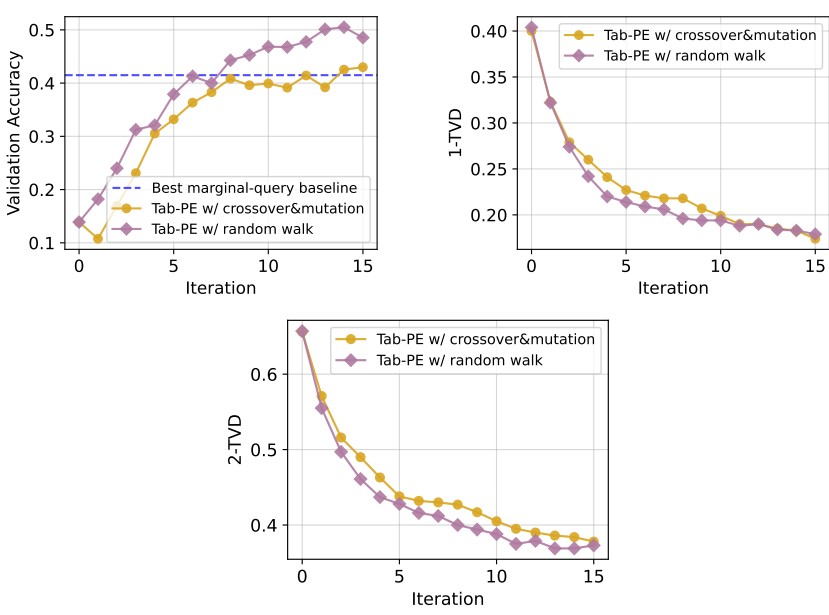

Figure 16: Comparing the proposed random-walk VARIATION_API with a genetic algorithm-based design.

## C.6 HYPERPARAMETER SENSITIVITY ANALYSIS

For consistency, this section presents the results on the Artificial Characters dataset with $\epsilon = 1.0$. For the hyperparameter sensitivity analysis, we vary one hyperparameter while keeping the others fixed as the default ones presented in the implementation if not specified.

**Number of synthetic samples** The smaller number of samples, the counts $hist$ are larger, which cause the noise to be less significant. However, too few samples may not be able to represent all high-dimensional correlations. If the number of samples is too large, the noise has more impact and can change the order of sample rankings. Figure 17 presents the performance of Tab-PE while varying the number of synthetic samples. At $\epsilon = 1.0$, 10% and 20% of the size of the private dataset achieve the best performance.

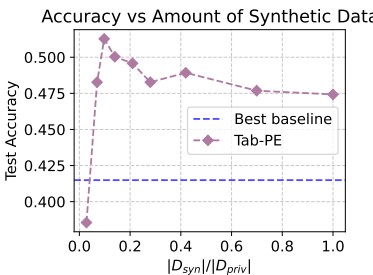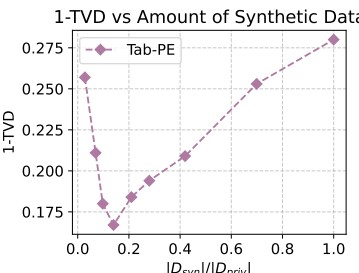

Figure 17: Tab-PE performance while varying the number of synthetic samples $\mathcal{D}_{\text{syn}}$.

**Number of iterations** A larger number of iterations $T$ allows more refinement of the synthetic data, but also leads to a larger noise scale $\sigma$. Figure 18 presents the performance of Tab-PE under different settings of the number of iterations $T$. Tab-PE needs around 15-20 iterations to achieve good performance.

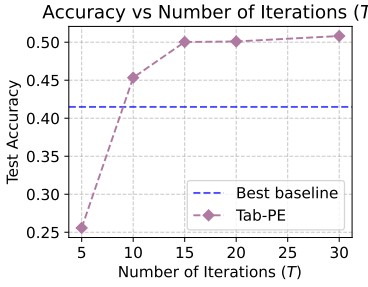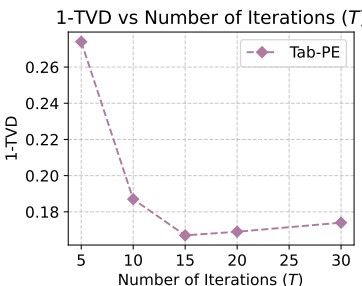

Figure 18: Tab-PE performance while varying the number of iterations $T$.

**Number of sampling iterations** The number of sampling iterations $T_{\text{sampling}}$ controls how many times we employ the sampling-with-replacement strategy. Figure 19 presents the performance of Tab-PE under different configurations of $T_{\text{sampling}}$. Generally, combining both sampling and ranking (i.e., $0 < T_{\text{sampling}} < T$) yields better performance than only ranking (i.e., $T_{\text{sampling}} = 0$) or only sampling (i.e., $T_{\text{sampling}} = T$). The best performance is achieved when using 20-40% of iterations for the sampling strategy.

**Mutation rate initial value** $\mu_{\text{init}}$ This parameter controls the noise level in the random walk strategy. A larger mutation rate enables more exploration, but also makes it harder for local refinement. Figure 20 presents the performance of Tab-PE with various values of $\mu_{\text{init}}$. A moderate value around 0.5-0.8 provides the best utility.

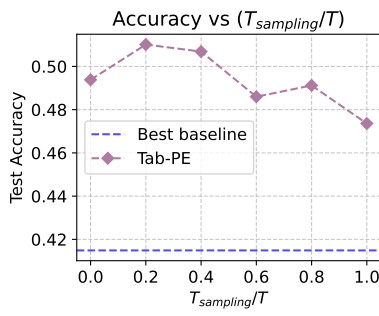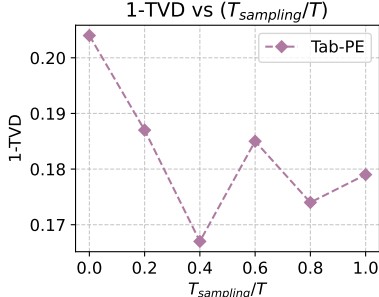

Figure 19: Tab-PE performance while varying the number of sampling iterations $T_{\text{sampling}}$.

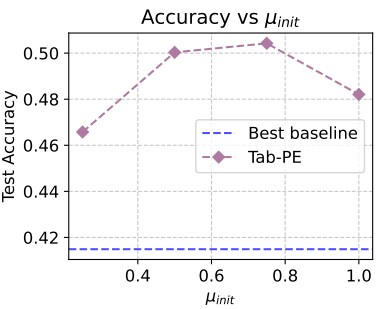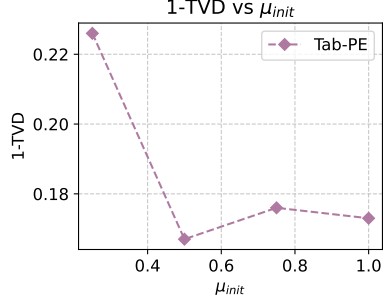

Figure 20: Tab-PE performance while varying the mutation initial rate $\mu_{init}$ in `VARIATION_API`.

**Decay factor** $\gamma$  This parameter controls how fast the mutation rate decays. A smaller $\gamma$ leads to a faster decay. A value at 1.0 is equal to a linear decay. Figure 21 presents the performance of Tab-PE with different settings of $\gamma$. A value around 0.5-0.75 achieves the best performance and outperforms the linear decay.

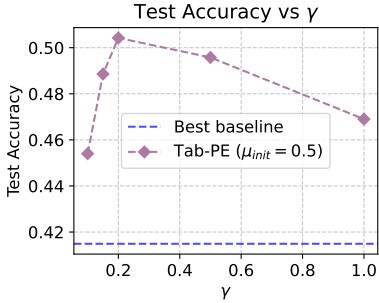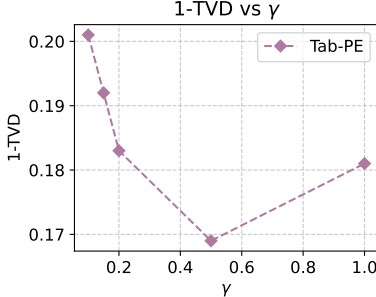

Figure 21: Tab-PE performance while varying $\gamma$ in the mutation rate schedule decay.

**Hyperparameter search**  In Figure 22, we explore $144$ settings of hyperparameters for the second stage (Tab-PE with ranking), the hyperparameters are chosen from the following sets: number of iterations (epochs) $\in \{15, 20, 30, 50\}$, num_samples $\in \{2k, 5k, 10k\}$, variation degree ($m = $ num_variations) $\in \{3, 5, 7\}$, and mutation rate initial value ($\mu_{init}$) $\in \{0.15, 0.25, 0.35, 0.5\}$. The mutation rate in this experiment is set by a linear decay schedule. Note that from the figure, smaller values of all hyperparameters generally do better. This inspires us to employ the polynomial decay schedule for the mutation rate, which enables a larger mutation rate at early iterations and a smaller mutation rate at later iterations.

**Hyperparameter configuration across privacy settings**  In Figure 23 for $\epsilon = 1.0$ (left-most column) we order all $144$ hyperparameters according to their achieved accuracy. $0$ corresponds to the

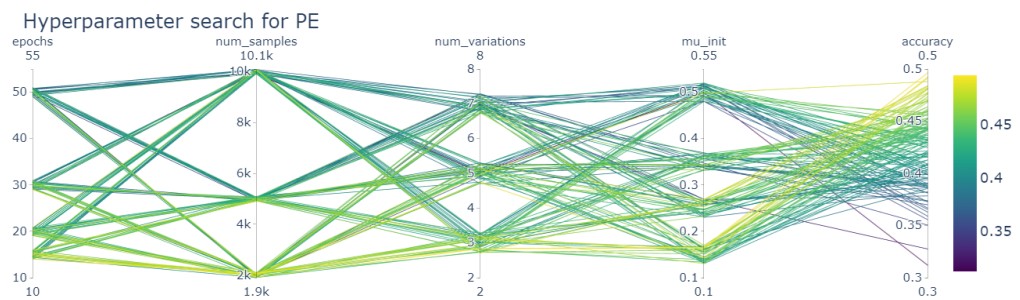

Figure 22: Hyperparameter search for Tab-PE on the Artificial Characters dataset for $\epsilon = 1.0$.

best setting of hyperparameters, 1 corresponds to the second best setting, and so on. The lines are colored according to their performance on $\epsilon = 1.0$. The same line corresponds to the same setting of hyperparameters. We note that the same hyperparameters that are good for $\epsilon = 1.0$ are also good for $\epsilon = 3.0$ and $\epsilon = 10.0$.

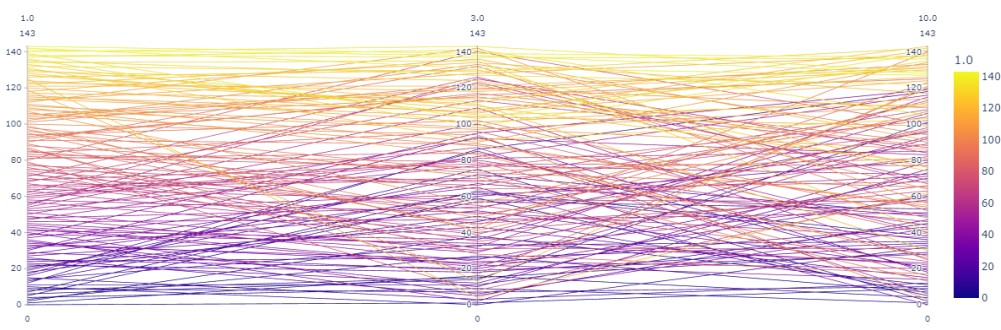

Figure 23: Ordering of the best hyperparameters for $\epsilon = 1.0$, $\epsilon = 3.0$ and $\epsilon = 10.0$.

### C.7 OVERSAMPLING STUDY

Following the simple recipe from PrivGSD (Liu et al., 2023), we conduct oversampling by randomly duplicating the samples. Figure 24 shows that the performance does not change significantly by oversampling.

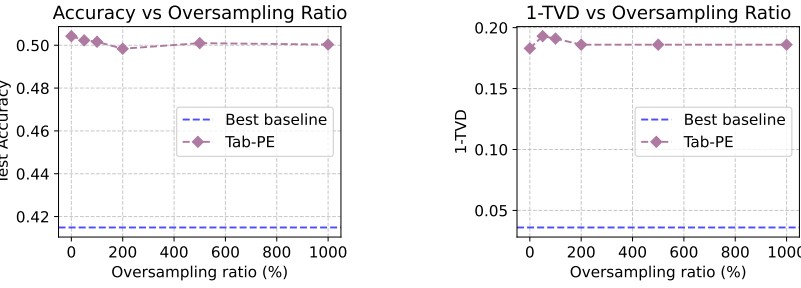

Figure 24: Tab-PE performance while enhancing by oversampling.

## C.8 NOISY CLASS DISTRIBUTION

We remove the assumption that the class distribution is public. Instead, we spend a bit of privacy budget to estimate the class distribution. To simplify, we spend $\epsilon_{\text{count}}$ out of $\epsilon$ for this estimation. Let $N_c$ is the count vector where $N_c[i]$ corresponds to the number of samples of class $i$. Since each sample is only counted once, the sensitivity of this counting process is 1. To achieve $(\epsilon_{\text{count}}, \delta)$-DP, we simply add noise, drawn from $\mathcal{N}(0, \sigma^2)$ to each count, where the noise multiplier is calculated by the analytic Gaussian Mechanism (Balle & Wang, 2018). In practice, our implementation uses the `diffprivlib` library to calculate this noise scale. We conduct an experiment spending 0.02 out of 1.0 to privately estimate the class counts. Figure 25 presents the results of this experiments. Overall, the performance does not change much with the assumption that the class distribution is publicly available.

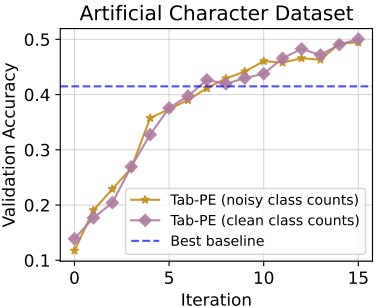 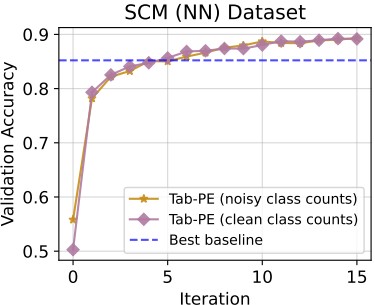

Figure 25: Tab-PE performance with noisy and clean class counts.

# D LIMITATIONS & FUTURE WORK

Although the results are promising, there are still limitations. First, while Tab-PE consistently outperforms the baselines in capturing high-order correlations with better ML utilities, it underperforms on fidelity metrics (i.e., TVD), which primarily reflects low-order statistics. Second, the gap between Tab-PE and the upper bound (non-private) remains large. This gap is significantly larger than the current gaps of datasets with low-order correlations. Therefore, there is still room for further improvements. Third, Tab-PE currently implements a basic distance function on raw attribute scaled values without any embedding or attribute weighting. This can suffer in extreme cases where the data is sparse and most of the attributes are irrelevant. While embedding in the image and text domains can be achieved by pretrained foundation models, tabular data is very challenging with strong distribution shift across datasets. We leave these for future work. Additionally, we only explored simple designs of the Private Evolution APIs. More advanced APIs may further boost the performance.

