# OpenReview forum: "Differentially Private Synthetic Data via APIs 4: Tabular Data"
_ICLR.cc/2026/Conference — Submitted to ICLR 2026_

### Official Review · Reviewer_9tC8 · 2025-10-16

**Soundness:** 1
**Presentation:** 2
**Contribution:** 1
**Rating:** 2
**Confidence:** 5

**Summary:**

This paper introduces Tab-PE, a differentially private data synthesis algorithm that adapts the Private Evolution (PE) framework to the tabular domain. The method iteratively perturbs and selects synthetic samples through a DP scoring mechanism based on nearest-neighbor histograms, aiming to capture high-order correlations between attributes without relying on query workloads or model-based approaches. The authors claim that Tab-PE achieves better downstream ML performance than prior baselines under comparable privacy budgets while being significantly faster and simpler to implement.

**Strengths:**

- Tab-PE is computationally lightweight, requires no model training, and scales efficiently to medium-sized tabular datasets, offering a practical alternative to query- or model-based DP synthesizers.

- Tab-PE achieves good downstream accuracy and privacy-utility trade-offs under moderate $\varepsilon$, outperforming several baselines.

**Weaknesses:**

- The paper overlooks major prior works that explicitly highlight or address high-order correlations in DP tabular synthesis, including [1], which identifies modeling higher-order correlations as a key open problem (at the end of Sec. 3.1), and Kamino [2], a constraint-aware DP synthesizer that directly tackles this challenge. **Failing to acknowledge or compare with these works reveals an incomplete understanding of the literature and weakens the positioning of the claimed contribution.**

- The authors never provide a concrete definition of “higher-order correlation”, in stark contrast to [1], which formalizes the concept through specific instances such as denial constraints and conditional functional dependencies. Here, the term is used loosely as a buzzword and assessed only through downstream ML performance, which is an unreliable proxy. It remains unclear whether the proposed Tab-PE method actually addresses the stated problem.

- Tab-PE performs substantially worse than AIM on low-order fidelity metrics (e.g., 1-way and 2-way TVD). If a method fails to preserve even low-order marginals, there is little reason to believe it can capture higher-order ones.

- The sampling procedure in PE (top-K) is inherently biased, leading to tail loss and mode concentration. Tab-PE inherits this limitation, which likely explains its poor 1-TVD and 2-TVD results compared to state-of-the-art approaches such as AIM.

- Tab-PE uses a single $\lambda$ to trade off categorical and numerical scales in DP-NN. This hyperparameter must be dataset-dependent and the results could be very sensitive to this choice. This is a weakness inherent to the proposed algorithm.

- The core algorithm is a direct application of PE with simple tabular adaptations (Gaussian perturbation for numerical attributes, resampling for categorical ones), reusing the same scoring and update loop from prior PE papers. It has very limited technical novelty and does not bring new insights to the field of DP tabular data synthesis.

- The paper deviates from established evaluation practices in DP tabular synthesis. Rather than benchmarking on standard datasets  widely used in prior work (e.g., Adult, Bank, Census, Hospital), it selects its own suite tailored to its method and highlights improvements. This practice hinders cross-paper comparison and undermines the credibility of the empirical claims. **A rigorous study should first have validated Tab-PE on recognized benchmarks before proposing new ones**.


[1] Hu, Yuzheng, et al. "Sok: Privacy-preserving data synthesis." 2024 IEEE Symposium on Security and Privacy (SP). IEEE, 2024.

[2] Ge, Chang, et al. "Kamino: Constraint-aware differentially private data synthesis." VLDB 2021

**Questions:**

Can you answer the following question: Is it possible for a synthetic tabular dataset to preserve high-order correlations present in the original dataset yet exhibit poor low-order fidelity?

---

> ### Author Response · Authors · 2025-11-21
> **Response to Reviewer #9tC8 (1/4)**
>
> We appreciate the reviewer for the thoughtful feedback and constructive comments. We response the main concerns and questions as follows:
>
> ### **Weakness 1: Related Work, Differences between Tab-PE scopes and Constraint-aware Methods**
> We thank the reviewer for mentioning the references. We would like to clarify the setting differences between our work and Kamino.
>
> **Different Settings**. The constraint-aware synthetic data generation methods such as Kamino address a different problem setting from Tab-PE. Kamino aims to enforce explicit denial constraints which are logical rules that certain feature value combinations are forbidden. While Tab-PE works on a standard synthetic data generation setting without constraints. Tab-PE aims to preserve the underlying high-order correlations that can be **complex, non-linear, and unknown**. Therefore, the constraint-aware setting can be complementary but not directly comparable to Tab-PE. We add this discussion and clarification in the revision.
>
> &nbsp;
>
> ### **Weakness 2: High-order correlations and ML Utility Relationship**
>
> To formalize high-order correlations, we can consider the mutual information of features and label. Let $X^{(k)} = (X_1, X_2, ..., X_k)$ is the set of $k$ relevant features and label $Y$. Without loss of generality, label $Y$ can be any feature in $X$. The mutual information $I(Y; X^{(k)})$ quantifies the total amount of information shared between $Y$ and the set of features $X^{(k)}$, denoted as:
>
> $I(Y; X^{(k)}) = H(Y) - H(Y|X^{(k)})$,
>
> where $H(Y)$ is the entropy of $Y$ and $H(Y|X^{(k)})$ is the conditional entropy of $Y$ given $X^{(k)}$.
>
> &nbsp;
>
> **Definition 1** (k+1)-way correlation: We say that there exists a (k+1)-way correlation between feature set $X^{(k)}$ and label $Y$ if the following condition holds:
>
> $I(Y; X^{(k)}) - \text{max}_{i \in \{1,..k\}}{\text{ }I(Y; X^{(k)} \setminus \{X_i\} }) \geq \Delta$
>
> This indicates that the full set $X^{(k)}$ provides at least $\Delta$ more mutual information about $Y$ than the any subset obtained by removing a single feature.
>
> &nbsp;
>
> **Exponential scaling**. However, in practice, finding high-order mutual information faces two sources of exponential growth with respect to the order $k$. First, the number of subset that need to be examined to detect $k$-way correlations grows combinatorially. For instance, a dataset with 15 features, checking for 4-way correlations requires computing all $\binom{15}{4} = 1365$ possible feature combinations. Second, each mutual information calculation itself also faces the curse of dimensionality due to the joint distribution estimation of multiple features. As a result, directly determining high-order correlations via mutual information is computationally expensive.
>
> **ML Utility is a good and practical indicator**. To overcome this challenge, we constraint the depth of decision trees as a proxy to capture the order of correlations. A tree of depth $k$ can represent interactions a long root-to-leaf path involving up to $k$ features. Training the ML models can effectively identify and leverage the most relevant high-order correlations for the prediction task. Additionally, ML models are fairly robust with noise and can automatically ignore irrelevant features. Therefore, by constraining the depth of decision trees, we can effectively find the order of correlations without exhaustively computing the mutual information of all feature combinations. Then, to quantify how well the synthetic data preserves high-order correlations, a very deep ML model (e.g., TabICL) is able to exploit all existing high-order correlations in the synthetic data. Thus, the resulting ML utility can serve as a practical and effective indicator of the preservation of high-order correlations in synthetic data.
>
> **Empirical Calculation**. Considering the stress test (XOR) data with two features $X_1, X_2$. We discretize the features into 20 bins. We calculate the mutual information using python `dit` package [1]. In this dataset, there is only a single correlation between $Y$ and $\{X_1, X_2\}$. The below table presents the mutual information values of different combinations. The mutual information $I(Y;X_1,X_2)$ is significantly larger than the ones of lower-order combinations. This indicates a 3-way correlation between $X_1, X_2$ and $Y$. This correlation also is well captured by ML models as presented in the paper.
>
> | Combination  | Value   |
> |----------|--------|
> | $I(X_1; X_2)$  | 0.0048  |
> | $I(Y;X_1)$   |  0.0004 |
> | $I(Y;X_2)$   |  0.0003 |
> | $I(Y;X_1,X_2)$   |  0.8970 |
>
>
> ---
> [1] https://github.com/dit/dit

---

> ### Author Response · Authors · 2025-11-21
> **Response to Reviewer #9tC8 (2/4)**
>
> ### **Weakness 3: Lower TVD fidelity but better high-order correlation preservation**
> The low-order fidelity metrics do not reveal the high-order interactions. Let us consider the simple example of the stress test (XOR) dataset with two features, which are visualized in Figure 9. In this simulation dataset, there is only one single correlation injected, regarding the dependence of label on the two features. The TVD fidelity metrics require the features to be distributed similarly to the private data, i.e., uniformly in this case, but it does not guarantee that the label values will be preserved correctly.
>
> Here we give an example to show that **the TVD metrics do not reveal the preserved high-order correlation**. Let us consider the stress test (XOR) dataset with three features. In this simulation dataset, there is only one *single* correlation injected: the dependence of label on the features. Achieving strong 1-TVD fidelity requires the features in the synthetic data to be distributed similarly to the private data (uniform in this case). However, this does not guarantee that the labels are assigned correctly according to the underlying high-order dependency.
>
> Because the underlying function is known, so we can directly compute the raw probability that a sample's label is correctly assigned given the feature values:
>
> $$
> P(Y=1 \mid (X_1>0 \land X_2<0 \land X_3<0)
>         \cup (X_1<0 \land X_2>0 \land X_3<0)
>         \cup \dots )
> $$
>
> The following table presents the TVD fidelity metrics of AIM and Tab-PE. Although Tab-PE has higher TVD errors, 91% of Tab-PE samples have the correct label, reflecting that it preserves the true high-order interaction. In contrast, AIM fails to capture this dependency. Although we configure AIM with the desired degree hyperparameter (i.e., 4), its dynamic query selection mechanism struggles to identify the right high-order marginals under DP noise and the large search space of 1–4-way queries.
>
> | Method  | 1-TVD  | 2-TVD  | Prediction Acc | $P(Y=1 \| X1,X2,X3)$ |
> |----------|--------|--------|----------------|-------------------|
> | AIM  | 0.006  | 0.035  | 0.510         | 0.502             |
> | Tab-PE   | 0.036  | 0.143  | 0.940          | 0.910             |
>
> Finally, the raw probability naturally aligns with downstream ML utility which highlights that the downstream performance is a practical and reliable indicator of preserved high-order correlations.

---

> ### Author Response · Authors · 2025-11-21
> **Response to Reviewer #9tC8 (3/4)**
>
> ### **Weakness 4: PE weaknesses**
> We appreciate the reviewer for mentioning the potential weaknesses of the general Private Evolution framework regarding the lower performance on TVD fidelity metrics. We will discuss these limitations in the revised version and explore potential solutions in future work.
>
> Although we acknowledge the importance of fidelity metrics which the research community has long focused on, we would like to emphasize that **our focus -- preserving high-order interactions is also crucial to enable many practical applications**. Here we give a few examples.
>
> &nbsp;
>
> **1. Classification Utility**: Not only Tab-PE offers better classification utility than the synthetic data baselines as presented in the paper. We have recently found ***the classifier trained on synthetic data generated by Tab-PE also significantly outperform the SOTA *direct* DP training classifiers in the context of high-order correlations***.
>
> Table 1. DP Classification Performance (epsilon = 1.0) on Artificial Characters dataset (higher is better)
>
> | Model          | Test Accuracy | Test F1 Score |
> |----------------|---------------|---------------|
> | DP Naive Bayes | 0.231 | 0.176 |
> | DP Decision Tree | 0.246 | 0.138 |
> | DP GBDT [1]    | 0.224 | 0.210|
> | Tab-PE + Decision Tree | 0.403 | 0.394 |
> | Tab-PE + TabICL|0.504 | 0.500|
> | Tab-PE + TabPFN|0.511 | 0.501|
>
> It is worth noting that for low-order datasets, the direct DP training methods still outperform classifiers trained on synthetic data generation. In particular, DP Gradient Boosted Decision Tree (GBDT) achieves 83.61% and 94.19% accuracy on Adult and Breast Cancer, compared to the best classifiers trained on synthetic data at 83.31% and 92.17%. This result indicates that **the existing SOTA methods for direct DP classification training performs well for low-order datasets but struggle to capture high-order correlations**.
>
> &nbsp;
>
> **2. Interpretability**: Preserving high-order correlations leads to better performance for interpretability tasks. The following table presents the feature importance calculated using SHAP values [2] of classifiers trained on synthetic data.
>
> Table 2. SHAP Feature Importance for Class 0 (values closer to the ones by private data are better)
> | Dataset: Artificial Characters | Feature 1 | Feature 2 | Feature 3 | Feature 4 | Feature 5 | Feature 6 | Feature 7 |
> |-----------------------------|----|----|----|----|----|----|----|
> | Private Data                | 0.559 | 0.191 | 0.588 | 0.366 | 0.473 | 0.139 | 0.404 |
> | PrivGSD (best accuracy baseline)     | 0.126 | 0.099 | 0.093 | 0.157 | 0.153 | 0.265 | 0.204 |
> | AIM | 0.023 | 0.010 | 0.003 | 0.014 | 0.006 | **0.079** | 0.148 |
> | Tab-PE                      | **0.626** | **0.279** | **0.473** | **0.415** | **0.429** | 0.517 | **0.513** |
>
> &nbsp;
>
> Table 3. SHAP Feature Importance for Class 1
> | Dataset: Artificial Characters | Feature 1 | Feature 2 | Feature 3 | Feature 4 | Feature 5 | Feature 6 | Feature 7 |
> |-----------------------------|----|----|----|----|----|----|----|
> | Private Data                | 0.523 | 0.244 | 0.992 | 0.353 | 0.180 | 0.347 | 0.946 |
> | PrivGSD (best accuracy baseline)     | **0.376** | 0.035 | 0.356 | 0.158 | **0.204** | 0.165 | 0.463 |
> | AIM | 0.090 | 0.005 | 0.005 | 0.005 | 0.006 | 0.031 | 0.277 |
> | Tab-PE                      | 0.763 | **0.127** | **0.622** | **0.540** | 0.487 | **0.326** | **0.899** |
>
> &nbsp;
>
> **3. Data Imputation**: Tab-PE offers better predictive performance not only for the label column but also for missing feature values. The following table presents the imputation accuracy using models trained on synthetic data by different methods.
>
> Table 4. Feature Imputation Accuracy (higher is better)
> | Dataset: Artificial Characters | Overall | Feature 1 | Feature 2 | Feature 3 | Feature 4 | Feature 5 | Feature 6 | Feature 7 |
> |----------------|-----------|-----------|-----------|-----------|-----------|-----------|-----------|-----------|
> | PrivGSD (best accuracy baseline)    | 0.449 | **0.487** | 0.686 | 0.489 | 0.547 | 0.405 | 0.376 | 0.266 |
> | AIM | 0.335 | 0.466 | 0.617 | 0.354 | 0.433 | 0.142 | 0.271 | 0.201 |
> | Tab-PE | **0.495** | 0.481 | **0.700** | **0.562** | **0.568** | **0.472** | **0.416** | **0.317** |
>
> &nbsp;
>
> ---
>
> [1] Maddock, Samuel et al. "Federated Boosted Decision Trees with Differential Privacy." CCS, 2022
>
> [2] Scott M. Lundberg and Su-In Lee. A unified approach to interpreting model predictions. NeurIPS, 2017.

---

> ### Author Response · Authors · 2025-11-21
> **Response to Reviewer #9tC8 (4/4)**
>
> ### **Weakness 5: Analysis on $\lambda$***
> We thank the reviewer for pointing this missing analysis. We have conducted additional experiments to analyze the impact of $\lambda$. The following table presents the performance of Tab-PE with different $\lambda$ values on the Person Activity dataset under $\epsilon=1.0$.
>
> | $\lambda$  | Test AUC  | Test F1  | 1-TVD | 2-TVD |
> |----------|--------|--------|----------------|-------------------|
> | 0.0  |  0.510 |  0.266 | 0.043 | 0.194 |
> | 0.1   | 0.630  | 0.350  | 0.035          | 0.124             |
> | 0.3333   | 0.631  | 0.354  | 0.044          | 0.130             |
> | 1.0   | 0.630  | 0.348  | 0.044          | 0.133             |
>
> Without consider categorical features ($\lambda = 0$), the performance significantly drops. The ML utility is quite stable across different non-zero $\lambda$ values.
>
> &nbsp;
> ### **Weakness 6, 7: Why new benchmark and our novelty**
>
> First, we would like to clarify that we have evaluated on some well-known datasets in Appendix C.3.
>
> **Existing Benchmark Limitations** Although it is intuitive that the methods using marginal queries would face with the exponential increase in the number of queries at high orders, the DP synthetic tabular data generation methods have still been dominated by these marginal-based methods *for years since 2021*. Our analyses on the order of correlations (Introduction and Appendix B.1) provide new insights to explain why the widely-used datasets and benchmarks may not be sufficient to reveal the limitations of these methods. From that, we provide a novel stress test and new benchmark datasets that can better evaluate the ability of preserving high-order correlations.
>
> **Novelty** While *most* methods in the recent DP synthetic tabular data literature have been built around statistical queries and optimizing the error of query answers for years, our work takes a novel perspective by evolutionary. We demonstrate that our approach offers certain advantages in datasets with high-order correlations and extremely efficient in computation. Additionally, Private Evolution has been an emerging paradigm in synthetic data generation with many follow-up works in different data modalities (e.g., images [1], text [2]).  Redesigning and demonstrating its effectiveness in a popular modality such as tabular data is a nontrivial contribution. We believe the high-order perspective and the proposed method can inspire new research directions in the tabular field.
>
> ---
>
> [1] Lin, Zinan et al. "Differentially Private Synthetic Data via Foundation Model APIs 1: Images.", ICLR 2024. (60+ citations)
>
> [2] Xie, Chulin et al. "Differentially Private Synthetic Data via Foundation Model APIs 2: Text." ICML 2024 (60+ citations)

---

> > ### Comment · Reviewer_9tC8 · 2025-11-21
> >
> > I appreciate the authors’ detailed rebuttal and additional analyses. However, several core concerns remain insufficiently resolved, particularly regarding the scope and meaning of “high-order correlations” and how Tab-PE meaningfully preserves them in practice.
> >
> > - The rebuttal introduces a definition based on conditional mutual information with respect to an exogenous label column. This is one reasonable lens, yet it is only *one interpretation* among many and does not correspond to the standard notion of high-order statistical dependence used in prior work on DP tabular synthesis, which typically refers to large $k$-way marginals or constraints over arbitrary subsets of attributes, not exclusively label-conditioned interactions. Two issues stem from this:
> >
> >   - Ambiguity: If “high-order correlation” only refers to label-conditioned interactions, then the method is solving a narrower problem than suggested in the introduction (and throughout the paper). If it is intended to capture general higher-order dependencies (over any subset of variables), then "mutual information with label" is not an adequate proxy and should not be used as the definition.
> >   - Empirical mismatch: The rebuttal claims that Tab-PE preserves higher-order correlations while AIM does not. Yet Tab-PE exhibits significantly worse 1-TVD and 2-TVD on many datasets (often 5 to 10 times worse). It is difficult to reconcile this with a claim that Tab-PE preserves general $k$-way marginals for large $k$. Generally, if a method cannot match low-order marginals, it is unlikely to match high-order ones unless the claimed “high-order structure” refers to a very restricted class of dependencies.
> >
> > - The XOR example is a useful sanity check, but it is not clear how the insights translate to real datasets such as Artificial Characters or Person Activity. The rebuttal claims that Tab-PE captures high-order correlations in these datasets, yet does not specify what those correlations actually are, how they manifest in the data distribution, or why Tab-PE is particularly suited to recover them. Without identifying the underlying interactions, it is difficult to attribute the observed ML-utility gains to genuine high-order preservation rather than other factors such as inductive bias or reduced noise on specific attributes.
> >
> > - Finally, I encourage the authors to more explicitly credit the SoK paper for framing the study of high-order correlations in DP tabular synthesis. This strengthens the positioning of the current work rather than detracting from it. I also think it would be interesting to discuss the potential misalignment between fidelity and utility-based metrics.

---

> > > ### Author Response · Authors · 2025-11-27
> > > **Follow-Up Response to Reviewer 9tC8 (1/3)**
> > >
> > > We would like to thank the reviewer for their constructive and insightful suggestions.
> > >
> > > &nbsp;
> > >
> > > ## **Regarding the formal definition**
> > > We thank the reviewer for highlighting this point. We would like to clarify that we have mentioned in the previous comment that without loss of generality, label $Y$ can be any feature in $X$. We have also demonstrated the downstream impact of this by a data imputation task. To save the reviewer's time, here we copy the previous results:
> > >
> > > **Data Imputation**: Tab-PE offers better predictive performance not only for the label column but also for missing feature values. The following table presents the imputation accuracy using models trained on synthetic data by different methods.
> > >
> > > Table 4. Feature Imputation Accuracy (higher is better)
> > > | Dataset: Artificial Characters | Overall | Feature 1 | Feature 2 | Feature 3 | Feature 4 | Feature 5 | Feature 6 | Feature 7 |
> > > |----------------|-----------|-----------|-----------|-----------|-----------|-----------|-----------|-----------|
> > > | PrivGSD (best accuracy baseline)    | 0.449 | **0.487** | 0.686 | 0.489 | 0.547 | 0.405 | 0.376 | 0.266 |
> > > | AIM | 0.335 | 0.466 | 0.617 | 0.354 | 0.433 | 0.142 | 0.271 | 0.201 |
> > > | Tab-PE | **0.495** | 0.481 | **0.700** | **0.562** | **0.568** | **0.472** | **0.416** | **0.317** |
> > >
> > >
> > > &nbsp;
> > >
> > > ## **Analysis on TVD, JSD, Wasserstein distance**
> > > Thank you again for the great suggestions, to understand more about the high-order fidelity metrics, we conduct some analysis about TVD, JSD (Jensen-Shannon Divergence), and Wasserstein fidelity metrics at different orders and settings.
> > >
> > > For $k$-TVD and $k$-JSD, we first discretize the data into bins. Then, for each $k$-column combination, we measure the empirical joint frequencies in both private and synthetic datasets and convert them into probability mass functions by normalizing the counts. Subsequently, we compute the TVD or JSD between the joint distributions of the two datasets. We repeat the process for all $k$-column combinations and calculate the average.
> > >
> > > For the Wasserstein distance, all the features are scaled to [0, 1], so no feature dominates the distance calculation. The Wasserstein distance is computed directly on the full $k$-dimensional subsets for each $k$-column combination. The $k$-W is the average value of all combinations. The pairwise cost matrix is built using the standard Euclidean (l2) distance.
> > >
> > > Please refer to the tables for the detailed results. We summarize the key observations as follows:
> > > - The choice of discretization bins can impact to TVD and JSD significantly, especially on low-order ones, depending on the dataset. Wasserstein distance does not require discretization.
> > > - TVD and JSD are biased toward the synthetic data size. In the paper, AIM's TVD is measured with the private data size, while the size of Tab-PE synthetic data is generally smaller. This makes the TVD results in the paper favor AIM, creating a large gap between Tab-PE and AIM.
> > > - Wasserstein distance is more robust to the synthetic data size.
> > > - The general trend is that AIM performs better on lower-order fidelity metrics, while Tab-PE performs better on higher-order fidelity metrics.
> > > - With the same size of dataset, the low-order fidelity metrics gap between AIM and Tab-PE reduces significantly.

---

> > > ### Author Response · Authors · 2025-11-27
> > > **Follow-Up Response to Reviewer 9tC8 (2/3)**
> > >
> > > ### **Artificial-Characters Dataset**
> > >
> > > &nbsp;
> > >
> > > | **10 bins**  | 1-TVD  | 2-TVD  | 3-TVD | 4-TVD | 5-TVD | 6-TVD | 7-TVD | 8-TVD |
> > > |----------|--------|--------|-------|-------|-------|-------|-------|-------|
> > > | AIM (private size) | **0.026881** | **0.137579** | **0.311344** | **0.535431** | 0.746572 | 0.882745 | 0.950171 | 0.979586|
> > > | AIM (Tab-PE's size) | 0.067188 | 0.238479 | 0.534491 | 0.801122 | 0.928104 | 0.975327 | 0.992710 | 0.998182 |
> > > | Tab-PE | 0.075617 | 0.183075 | 0.352743 | 0.553896 | **0.712036** | **0.812589** | **0.873387** | **0.911038** |
> > > | **20 bins** |
> > > | AIM (private size)        | **0.035445** | **0.176709** | **0.417548** | **0.698039** | **0.876618** | **0.954169** | 0.984497 | 0.995805 |
> > > | AIM (Tab-PE's size) | 0.067188 | 0.238479 | 0.534491 | 0.801122 | 0.928104 | 0.975327 | 0.992710 | 0.998182 |
> > > | Tab-PE | 0.182792 | 0.379879 | 0.638086 | 0.832459 | 0.920391 | 0.957522 | **0.975403** | **0.984717**|
> > >
> > > &nbsp;
> > >
> > > | **10 bins**  | 1-JSD  | 2-JSD  | 3-JSD | 4-JSD | 5-JSD | 6-JSD | 7-JSD | 8-JSD |
> > > |----------|--------|--------|-------|-------|-------|-------|-------|-------|
> > > | AIM (private size) | **0.002002** | **0.028840** | **0.110675** | **0.262665** | 0.440088 | 0.569201 | 0.637855 | 0.670122|
> > > | AIM (Tab-PE's size) | 0.004210 | 0.041247 | 0.162235 | 0.369182 | 0.540581 | 0.629366 | 0.667776 | 0.684533 |
> > > | Tab-PE | 0.009173 | 0.041745 | 0.136954 | 0.288252 | **0.423094** | **0.514129** | **0.570781** | **0.607037** |
> > > | **20 bins** |
> > > | AIM (private size) | **0.004184** | **0.047223** | **0.192978** | **0.421651** | **0.578038** | **0.648126** | **0.676540** | 0.688806 |
> > > | AIM (Tab-PE's size) | 0.006742 | 0.071385 | 0.281982 | 0.510184 | 0.625681 | 0.669483 | 0.686010 | 0.691374 |
> > > | Tab-PE | 0.048714 | 0.148365 | 0.363162 | 0.539298 | 0.620076 | 0.655000 | 0.672700 | **0.683023** |
> > >
> > > &nbsp;
> > >
> > > | Wasserstein  | 1-W  | 2-W  | 3-W | 4-W | 5-W | 6-W | 7-W | 8-W |
> > > |----------|--------|--------|-------|-------|-------|-------|-------|-------|
> > > | AIM (private size) | **0.009605** | **0.037011** | **0.078604** | 0.131570 | 0.188566 | 0.245150 | 0.299776 | 0.351410 |
> > > | AIM (Tab-PE's size) | 0.014817 | 0.044442 | 0.087082 | 0.140160 | 0.197000 | 0.253496 | 0.307995 | 0.359972 |
> > > | Tab-PE | 0.037793 | 0.065208 | 0.095395 | **0.130072** | **0.164409** | **0.196297** | **0.225520** | **0.252635** |

---

> ### Author Response · Authors · 2025-11-27
> **Follow-Up Response to Reviewer 9tC8 (3/3)**
>
> ### **Person-Activity Dataset**
> Due to the computational cost, as this dataset is signficantly larger than Artificial-Characters, we only present the results of the same size synthetic data size.
>
> &nbsp;
>
> | **10 bins**  | 1-TVD  | 2-TVD  | 3-TVD | 4-TVD | 5-TVD | 6-TVD | 7-TVD | 8-TVD |
> |----------|--------|--------|-------|-------|-------|-------|-------|-------|
> | AIM (Tab-PE's size) | **0.012771** | **0.053379** | **0.143229** | **0.296869** | **0.488874** | 0.668335 | 0.809941 | 0.900310 |
> | PE      | 0.038448 | 0.090559 | 0.188484 | 0.345468 | 0.514707 | **0.652367** | **0.750849** | **0.814606** |
> | **20 bins**  |
> | AIM (Tab-PE's size) | **0.017450** | **0.094118** | **0.273800** | **0.536738** | **0.761298** | 0.897294 | 0.964074 | 0.989780 |
> | PE      | 0.044399 | 0.132814 | 0.324634 | 0.573495 | 0.762776 | **0.868473** | **0.922113** | **0.949755** |
>
> &nbsp;
>
> | **10 bins** | 1-JSD    | 2-JSD    | 3-JSD    | 4-JSD    | 5-JSD    | 6-JSD    | 7-JSD    | 8-JSD    |
> |---------|----------|----------|----------|----------|----------|----------|----------|----------|
> | AIM (Tab-PE's size)| **0.000177** | **0.005092** | **0.027778** | **0.105727** | **0.237561** | 0.378928 | 0.500337 | 0.585041 |
> | PE      | 0.001500 | 0.012480 | 0.048344 | 0.138716 | 0.261444 | **0.372463** | **0.455126** | **0.510942** |
> | **20 bins**  |
> | AIM (Tab-PE's size)| **0.000315** | **0.013355** | **0.093527** | **0.273425** | **0.457828**     | 0.582623     | 0.649735     | 0.678809     |
> | PE      | 0.002247 | 0.024329 | 0.121237 | 0.299010 | 0.459269 | **0.558621** | **0.611397** | **0.639753** |
>
>
> &nbsp;
>
> | Wasserstein  | 1-W  | 2-W  | 3-W | 4-W | 5-W | 6-W | 7-W | 8-W |
> |----------|--------|--------|-------|-------|-------|-------|-------|-------|
> | AIM (Tab-PE's size)| **0.006365** | **0.015632** | **0.034485** | **0.062834** | 0.096774 | 0.133273 | 0.171545 | 0.210860 |
> | Tab-PE | 0.006947 | 0.016937 | 0.035587 | 0.062057 | **0.091398** | **0.120872** | **0.149365** |  **0.177133** |
>
>
> &nbsp;
>
> &nbsp;
>
> ### **(Simulation dataset) SCN-NN**
>
> &nbsp;
>
> | Wasserstein  | 1-W  | 2-W  | 3-W | 4-W | 5-W | 6-W | 7-W | 8-W |
> |----------|--------|--------|-------|-------|-------|-------|-------|-------|
> | AIM (Tab-PE's size)| 0.019650 | 0.034025 | 0.055341 | 0.084672 | 0.118352 | 0.153289 | 0.187920 | 0.221675 |
> | Tab-PE | **0.017594** | **0.032290** | **0.054305** | **0.084385** | **0.117958** | **0.151657** | **0.184001** | **0.214603** |
>
>
> &nbsp;
> &nbsp;
>
>
> ## Related work
> We thank the reviewer again for bringing this up. We will carefully discuss the similarities and differences of the mentioned references and ours. Despite the differences in problem setups that we have discussed in the previous comments, the SoK paper points out the weaknesses of low-order marginals and **consider this as an open problem**. Tab-PE is directly aligned with this perspective. We attach the pertinent text from the SoK paper below.
> ```
> Open Problems.
>  • Fine-grained structures. State-of-the-art algorithms for differentially private tabular data synthesis primarily rely on low-order marginals, which may not adequately capture the high-order correlations between attributes (e.g., denial constraints [142] and (conditional) functional dependencies [143, 144]). Integrating these fine-grained structures into the design of algorithms holds the potential to enhance data utility, a prominent example being Kamino [145].
> ```
>
> &nbsp;
> &nbsp;
>
> ---
> ---
>
> &nbsp;
>
>
> We greatly appreciate the reviewer's time and thoughtful suggestions. If any new questions arise, we are happy to provide further clarification.
>
>  &nbsp;
>
>
> Best regards,
>
> The authors

---

### Official Review · Reviewer_QPTQ · 2025-10-28

**Soundness:** 3
**Presentation:** 3
**Contribution:** 3
**Rating:** 6
**Confidence:** 4

**Summary:**

The paper introduces Tab-PE, a method for generating differentially private synthetic tabular data by adapting the popular Private Evolution (PE) framework to tabular data. The paper argues that existing SOTA tabular DP-SDG methods, like marginal-based approaches, struggle to capture high-order correlations due to their restricted simplicity in measuring low-dimensional marginals. They argue that Tab-PE is an effective alternative as it does not need to explicitly instantiate many high dimensional marginal queries. They show empirically Tab-PE outperforms on datasets with complex dependencies while being much faster than current SOTA marginal-based methods like AIM.

**Strengths:**

- The paper clearly identifies and empirically validates a meaningful limitation in existing DP-SDG algorithms namely, their inability to scale to many columns and their inability to capture high-order correlations.
- The paper is generally well organized, includes detailed algorithm descriptions and ablations, and the authors provide an anonymized code repository.
- The adaptation of the PE framework to tabular data without using LLMs/foundation models is conceptually straightforward yet shown to be very effective against SOTA approaches .

**Weaknesses:**

- The paper should more clearly delineate how Tab-PE differs from the closest related work. Namely the approach of Swanberg et al. and the PrivGSD method (see questions below).
- The presentation surrounding the notion of "APIs" is unclear/misleading since, as far as I can tell, the work does not use APIs in the traditional PE sense e.g., leveraging foundation models.
- Some ablations (e.g., replacing Tab-PE’s variation API with PrivGSD’s crossover/mutation) are relegated mostly to the appendix but would be valuable to feature more prominently in the main text to clarify differences with the closest related work.
- The evaluation focuses on datasets with only a few columns. Given the central claim about scalability and modeling high-order correlations, it would be more valuable to include experiments on higher-dimensional data where this benefit is much clearer.

**Questions:**

1. The general presentation around "APIs" is somewhat confusing and misleading. Why retain the "API" terminology if no external API calls or foundation models are actually used as in traditional PE?
2. Related to the above, more could be done to delineate this work from Swanberg et al. which seems the closest related work from a tabular PE perspective. Why have you chosen not to empirically compare to this approach?
3. It seems that PrivGSD is the closest related DP-SDG approach since it adapts the genetic algorithm approach to produce private tabular data which is closely related to Private Evolution. The specific differences between PrivGSD should be made much clearer in the main paper. Could you summarise the main differences?
4. The main argument for Tab-PE is that it retains high-order correlations and is much faster than competing methods like AIM. However, most of the benchmark datasets that are used have only a small number of columns. I would have preferred to have seen results that show Tab-PE on higher-dimensional datasets where the argument about capturing high-order correlation is usually much more necessary than on smaller datasets. Do you think Tab-PE can scale effectively to these scenarios?
5. Have you thought about lightweight hybrid approaches to extend Tab-PE with query-based methods (e.g., marginal information)? It seems to me a key weakness of Tab-PE is that it doesn’t capture 1-way or 2-way marginals as well as methods like AIM (which makes sense as AIM is explicitly trained to do so). However, it is fairly lightweight (from a DP perspective) to integrate marginal information e.g., using 1-way marginal information to initialize the Tab-PE approach (instead of random data).

---

> ### Author Response · Authors · 2025-11-21
> **# Response to Reviewer QPTQ**
>
> We thank the reviewer for the thoughtful feedback and constructive comments. We respond to the main concerns and questions as follows:
>
> ### **Weakness 1, Questions 2 & 3: Related Works**
>
> We thank the reviewer for noting the close related. Here we clarify the differences between Tab-PE and these methods.
>
> **Swanberg et al.** [1] is also a recent work that explores Private Evolution for tabular data. There are many differences but the most notable one is that they focus on leveraging LLM APIs and primarily optimizing for the low-order marginal queries. The work concludes that the method does not outperform the leading SOTA methods. Although the paper provides the great insights of the first exploration of Private Evolution for tabular data, but given the expensive cost of LLM API calls and the paper conclusions, we do not consider it as a strong baseline for comparison. In contrast, Tab-PE is designed with simple, efficient, and effective heuristic APIs without LLMs and marginal queries. We also demonstrate the advantages and success of Private Evolution.
>
> **Liu et al.** (PrivGSD) [2] is another evolutionary-based approach. PrivGSD first privately answer a set of statistical queries and then applies a genetic algorithm to optimize the workload error of these queries. The variation operations in PrivGSD work at dataset level, which is computationally expensive but required to calculate the fitness scores based on the query answers. In contrast, Tab-PE works at record level, which is significantly more efficient. Additionally, the evolution objective of PrivGSD and Tab-PE are different. PrivGSD selects dataset candidate based on the query answers, while Tab-PE selects records based on the pairwise distances.
>
> &nbsp;
>
> ### **Weakness 2 and Question 1: API notions**
> We thank the reviewer for pointing this out! As Private Evolution is emerging as a new paradigm for synthetic data generation with many follow-up works in different data modalities [3,4], these key notions can help readers who are familiar with previous Private Evolution works to better understand our method and connect Tab-PE to prior works. We will strongly consider your great suggestion to either clarify more or remove these notions to make the paper more accessible.
>
> &nbsp;
>
> ### **Weakness 3: Ablation/Analysis Presentation**
> We totally agree that these studies are important and insightful for Tab-PE's design choices. Due to the space limit, we had to move some ablation studies to the Appendix. We will move them back to the main paper in the revised version given one more extra page.
>
> &nbsp;
>
> ### **Weakness 4: High-dimensional settings**
> We appreciate the reviewer for pointing out this aspect. To simulate different number of features, we conduct an experiment reusing the SCM (NN) simulation setting. The below table presents the performance of Tab-PE and leading baselines. Generally, the performance of all methods drop as the problem getting more challenging and the gap between Tab-PE and the baselines become more significant.
>
> | # features        | Method | Test Acc | Test AUC | 1-TVD | 2-TVD |
> |------------------|--------|----------|----------|-------|-------|
> | 8 | PrivMRF | 0.849 | 0.919 | 0.036 |  0.081
> | 8 | AIM | 0.846 | 0.929 | 0.035 | 0.080 |
> | 8 | Tab-PE | 0.892 | 0.963 | 0.061 | 0.158 |
> | 20 | PrivMRF | 0.704 | 0.779 | 0.013 | 0.119 |
> | 20 | AIM | 0.701 | 0.783 | 0.007 | 0.117 |
> | 20 | Tab-PE | 0.777 | 0.864 | 0.133 | 0.253 |
>
> &nbsp;
>
> ### **Question 5: Hybrid approach**
> This is a great suggestion! We agree that to enhance the fidelity metrics we should integrate the marginal information. We will add this suggestion as potential future work in the revised version.
>
> ---
>
> [1] Swanberg, Marika et al. "Is API Access to LLMs Useful for Generating Private Synthetic Tabular Data?" ArXiv abs/2502.06555 (2025)
>
> [2] Liu, Terrance et al. "Generating Private Synthetic Data with Genetic Algorithms." ArXiv abs/2306.03257 (2023)
>
> [3] Lin, Zinan et al. "Differentially Private Synthetic Data via Foundation Model APIs 1: Images.", ICLR 2024. (60+ citations)
>
> [4] Xie, Chulin et al. "Differentially Private Synthetic Data via Foundation Model APIs 2: Text." ICML 2024 (60+ citations)

---

> > ### Comment · Reviewer_QPTQ · 2025-11-27
> >
> > I would like to thank the authors for their rebuttal and for clarifying the distinctions between their approach and the methods of Swanberg et al. and Liu et al. While the additional explanations are helpful, the new high-dimensional experiments remain unconvincing as 20 features is still low-dimensional. As other reviewers have noted, the current empirical evaluation does not clearly demonstrate that Tab-PE preserves high-order correlations to the extent claimed. As a result, I maintain my original score.

---

> > > ### Author Response · Authors · 2025-11-27
> > > **Follow-Up Response to Reviewer QPTQ (1/3)**
> > >
> > > We would like to thank the reviewer for their constructive and insightful comments.
> > >
> > > ## **High-dimensional settings**
> > > We thank the reviewer for the valuable feedback. As the  high-order and high-dimensional setting is inherently challenging. When the number of features becomes very large, even the non-private classifier struggles. In such cases, all DP methods tend to converge toward random guess performance, which do not provide very meaningful insights for comparison between DP synthetic data methods.
> > >
> > > ## **Analysis on TVD, JSD, Wasserstein distance**
> > > We appreciate the reviewer acknowledging the other discussions. To understand more about the high-order fidelity metrics, we conduct some analysis about TVD, JSD (Jensen-Shannon Divergence), and Wasserstein fidelity metrics at different orders and settings.
> > >
> > > For $k$-TVD and $k$-JSD, we first discretize the data into bins. Then, for each $k$-column combination, we measure the empirical joint frequencies in both private and synthetic datasets and convert them into probability mass functions by normalizing the counts. Subsequently, we compute the TVD or JSD between the joint distributions of the two datasets. We repeat the process for all $k$-column combinations and calculate the average.
> > >
> > > For the Wasserstein distance, all the features are scaled to [0, 1], so no feature dominates the distance calculation. The Wasserstein distance is computed directly on the full $k$-dimensional subsets for each $k$-column combination. The $k$-W is the average value of all combinations. The pairwise cost matrix is built using the standard Euclidean (l2) distance.
> > >
> > > Please refer to the tables for the detailed results. We summarize the key observations as follows:
> > > - The choice of discretization bins can impact to TVD and JSD significantly, especially on low-order ones, depending on the dataset. Wasserstein distance does not require discretization.
> > > - TVD and JSD are biased toward the synthetic data size. In the paper, AIM's TVD is measured with the private data size, while the size of Tab-PE synthetic data is generally smaller. This makes the TVD results in the paper favor AIM, creating a large gap between Tab-PE and AIM.
> > > - Wasserstein distance is more robust to the synthetic data size.
> > > - The general trend is that AIM performs better on lower-order fidelity metrics, while Tab-PE performs better on higher-order fidelity metrics.
> > > - With the same size of dataset, the low-order fidelity metrics gap between AIM and Tab-PE reduces significantly.

---

> > > ### Author Response · Authors · 2025-11-27
> > > **Follow-Up Response to Reviewer QPTQ (2/3)**
> > >
> > > ### **Artificial-Characters Dataset**
> > >
> > > &nbsp;
> > >
> > > | **10 bins**  | 1-TVD  | 2-TVD  | 3-TVD | 4-TVD | 5-TVD | 6-TVD | 7-TVD | 8-TVD |
> > > |----------|--------|--------|-------|-------|-------|-------|-------|-------|
> > > | AIM (private size) | **0.026881** | **0.137579** | **0.311344** | **0.535431** | 0.746572 | 0.882745 | 0.950171 | 0.979586|
> > > | AIM (Tab-PE's size) | 0.067188 | 0.238479 | 0.534491 | 0.801122 | 0.928104 | 0.975327 | 0.992710 | 0.998182 |
> > > | Tab-PE | 0.075617 | 0.183075 | 0.352743 | 0.553896 | **0.712036** | **0.812589** | **0.873387** | **0.911038** |
> > > | **20 bins** |
> > > | AIM (private size)        | **0.035445** | **0.176709** | **0.417548** | **0.698039** | **0.876618** | **0.954169** | 0.984497 | 0.995805 |
> > > | AIM (Tab-PE's size) | 0.067188 | 0.238479 | 0.534491 | 0.801122 | 0.928104 | 0.975327 | 0.992710 | 0.998182 |
> > > | Tab-PE | 0.182792 | 0.379879 | 0.638086 | 0.832459 | 0.920391 | 0.957522 | **0.975403** | **0.984717**|
> > >
> > > &nbsp;
> > >
> > > | **10 bins**  | 1-JSD  | 2-JSD  | 3-JSD | 4-JSD | 5-JSD | 6-JSD | 7-JSD | 8-JSD |
> > > |----------|--------|--------|-------|-------|-------|-------|-------|-------|
> > > | AIM (private size) | **0.002002** | **0.028840** | **0.110675** | **0.262665** | 0.440088 | 0.569201 | 0.637855 | 0.670122|
> > > | AIM (Tab-PE's size) | 0.004210 | 0.041247 | 0.162235 | 0.369182 | 0.540581 | 0.629366 | 0.667776 | 0.684533 |
> > > | Tab-PE | 0.009173 | 0.041745 | 0.136954 | 0.288252 | **0.423094** | **0.514129** | **0.570781** | **0.607037** |
> > > | **20 bins** |
> > > | AIM (private size) | **0.004184** | **0.047223** | **0.192978** | **0.421651** | **0.578038** | **0.648126** | **0.676540** | 0.688806 |
> > > | AIM (Tab-PE's size) | 0.006742 | 0.071385 | 0.281982 | 0.510184 | 0.625681 | 0.669483 | 0.686010 | 0.691374 |
> > > | Tab-PE | 0.048714 | 0.148365 | 0.363162 | 0.539298 | 0.620076 | 0.655000 | 0.672700 | **0.683023** |
> > >
> > > &nbsp;
> > >
> > > | Wasserstein  | 1-W  | 2-W  | 3-W | 4-W | 5-W | 6-W | 7-W | 8-W |
> > > |----------|--------|--------|-------|-------|-------|-------|-------|-------|
> > > | AIM (private size) | **0.009605** | **0.037011** | **0.078604** | 0.131570 | 0.188566 | 0.245150 | 0.299776 | 0.351410 |
> > > | AIM (Tab-PE's size) | 0.014817 | 0.044442 | 0.087082 | 0.140160 | 0.197000 | 0.253496 | 0.307995 | 0.359972 |
> > > | Tab-PE | 0.037793 | 0.065208 | 0.095395 | **0.130072** | **0.164409** | **0.196297** | **0.225520** | **0.252635** |

---

> > > ### Author Response · Authors · 2025-11-27
> > > **Follow-Up Response to Reviewer QPTQ (3/3)**
> > >
> > > ### **Person-Activity Dataset**
> > > Due to the computational cost, as this dataset is signficantly larger than Artificial-Characters, we only present the results of the same size synthetic data size.
> > >
> > > &nbsp;
> > >
> > > | **10 bins**  | 1-TVD  | 2-TVD  | 3-TVD | 4-TVD | 5-TVD | 6-TVD | 7-TVD | 8-TVD |
> > > |----------|--------|--------|-------|-------|-------|-------|-------|-------|
> > > | AIM (Tab-PE's size) | **0.012771** | **0.053379** | **0.143229** | **0.296869** | **0.488874** | 0.668335 | 0.809941 | 0.900310 |
> > > | PE      | 0.038448 | 0.090559 | 0.188484 | 0.345468 | 0.514707 | **0.652367** | **0.750849** | **0.814606** |
> > > | **20 bins**  |
> > > | AIM (Tab-PE's size) | **0.017450** | **0.094118** | **0.273800** | **0.536738** | **0.761298** | 0.897294 | 0.964074 | 0.989780 |
> > > | PE      | 0.044399 | 0.132814 | 0.324634 | 0.573495 | 0.762776 | **0.868473** | **0.922113** | **0.949755** |
> > >
> > > &nbsp;
> > >
> > > | **10 bins** | 1-JSD    | 2-JSD    | 3-JSD    | 4-JSD    | 5-JSD    | 6-JSD    | 7-JSD    | 8-JSD    |
> > > |---------|----------|----------|----------|----------|----------|----------|----------|----------|
> > > | AIM (Tab-PE's size)| **0.000177** | **0.005092** | **0.027778** | **0.105727** | **0.237561** | 0.378928 | 0.500337 | 0.585041 |
> > > | PE      | 0.001500 | 0.012480 | 0.048344 | 0.138716 | 0.261444 | **0.372463** | **0.455126** | **0.510942** |
> > > | **20 bins**  |
> > > | AIM (Tab-PE's size)| **0.000315** | **0.013355** | **0.093527** | **0.273425** | **0.457828**     | 0.582623     | 0.649735     | 0.678809     |
> > > | PE      | 0.002247 | 0.024329 | 0.121237 | 0.299010 | 0.459269 | **0.558621** | **0.611397** | **0.639753** |
> > >
> > >
> > > &nbsp;
> > >
> > > | Wasserstein  | 1-W  | 2-W  | 3-W | 4-W | 5-W | 6-W | 7-W | 8-W |
> > > |----------|--------|--------|-------|-------|-------|-------|-------|-------|
> > > | AIM (Tab-PE's size)| **0.006365** | **0.015632** | **0.034485** | **0.062834** | 0.096774 | 0.133273 | 0.171545 | 0.210860 |
> > > | Tab-PE | 0.006947 | 0.016937 | 0.035587 | 0.062057 | **0.091398** | **0.120872** | **0.149365** |  **0.177133** |
> > >
> > >
> > > &nbsp;
> > >
> > > &nbsp;
> > >
> > > ### **(Simulation dataset) SCN-NN**
> > >
> > > &nbsp;
> > >
> > > | Wasserstein  | 1-W  | 2-W  | 3-W | 4-W | 5-W | 6-W | 7-W | 8-W |
> > > |----------|--------|--------|-------|-------|-------|-------|-------|-------|
> > > | AIM (Tab-PE's size)| 0.019650 | 0.034025 | 0.055341 | 0.084672 | 0.118352 | 0.153289 | 0.187920 | 0.221675 |
> > > | Tab-PE | **0.017594** | **0.032290** | **0.054305** | **0.084385** | **0.117958** | **0.151657** | **0.184001** | **0.214603** |
> > >
> > > &nbsp;
> > >
> > > &nbsp;
> > >
> > > ---
> > > ---
> > >
> > > &nbsp;
> > >
> > >
> > > We greatly appreciate the reviewer's time and thoughtful suggestions. If any new questions arise, we are happy to provide further clarification.
> > >
> > >  &nbsp;
> > >
> > >
> > > Best regards,
> > >
> > > The authors

---

### Official Review · Reviewer_HPHj · 2025-11-01

**Soundness:** 4
**Presentation:** 3
**Contribution:** 3
**Rating:** 4
**Confidence:** 5

**Summary:**

This paper proposes a new tabular data synthesis algorithm that leverages the recent Private Evaluation (PE) technique. The method uses an API to generate candidate synthetic records and employs private histograms to iteratively select synthetic records that are more similar to the private data. Extensive experiments demonstrate the effectiveness of the proposed method across various scenarios, including low-order correlations, high-order correlations, and synthetic datasets.

**Strengths:**

1. The use of a simple XOR experiment to illustrate the weaknesses of existing marginal-based synthesis algorithms is interesting and well-motivated.

2. Employing a simple synthesis API (e.g., directly perturbing data through random selection or Gaussian noise) instead of relying on LLMs is an intriguing choice that still achieves strong performance.

3. The proposed method is simple, easy to implement, and computationally efficient.

**Weaknesses:**

1. Although the paper is overall interesting and well-written, I am concerned about the performance of Tab-PE. As shown in Table 1, the overall improvement over baseline methods is modest (at most around 10%), while the fidelity (even for simple one-way marginals) is significantly worse (up to 10 times worse on the Person Activity dataset). Given that low-order marginals are a crucial fidelity metric and have many applications (e.g., point/range queries) in tabular data, this represents a critical weakness. The paper should discuss this issue more thoroughly, especially regarding how Tab-PE should be used in practice.

2. While it is interesting to distinguish between low-correlation and high-correlation datasets, the performance differences of Tab-PE across these datasets are not particularly significant. In addition, Table 5 (fidelity evaluation on the Breast Cancer dataset) lacks bold or underlined highlights; this can be either a typo or a mistake in the results. Either way, Table 5 again shows that the advantage of Tab-PE over marginal-based methods lies mainly in ML prediction and embedding metrics, which is unsurprising and has already been validated by many prior studies.

3. Tab-PE relies on several pieces of so-called "public" information about the private tabular data, such as the domains of continuous and categorical values and class distributions. While the authors provide some justification for why these can be treated as public, directly using such information weakens the end-to-end privacy guarantee of Tab-PE and may not be practical when these statistics are unavailable. The authors should discuss these assumptions more carefully and provide a more rigorous privacy analysis.

**Questions:**

Q1. Given the relatively modest performance gains (and in fidelity, worse results than baseline methods), why do the authors claim that Tab-PE represents a new paradigm for tabular data synthesis? Under what conditions should Tab-PE be preferred over marginal-based methods?

Q2. Why does the performance of Tab-PE appear similar on both high-correlation datasets (Table 1) and low-correlation datasets (Table 5)? This similarity seems to contradict the motivation for distinguishing between these two types of datasets.

Q3. Why are certain statistics treated as public information, given that such assumptions could risk leaking private information?

**Details Of Ethics Concerns:**

This work studies the differentially private tabular data synthesis on public datasets and synthetic datasets.

---

> ### Author Response · Authors · 2025-11-21
> **# Response to Reviewer HPHj (1/2)**
>
> We thank the reviewer for the thoughtful feedback and constructive comments. We response the main concerns and questions as follows:
>
> ### **Weakness 1: Practical Uses**
>
> We appreciate the reviewer for raising this important aspect. We acknowledge the lower performance of Tab-PE on low-order fidelity metrics compared to baselines that *explicitly* optimize for low-order marginals. Improving workload error and low-order marginals has been extensively explored by the community and methods directly optimizing for them have been dominated the field for years. However, here we would like to emphasize that, **preserving high-order interactions is also crucial for many practical applications**.
>
> &nbsp;
>
> **1. Classification Utility**: Not only Tab-PE offers better classification utility than the synthetic data baselines as presented in the paper. We have recently found ***the classifier trained on synthetic data generated by Tab-PE also significantly outperform the SOTA *direct* DP training classifiers in the context of high-order correlations***.
>
> Table 1. DP Classification Performance (epsilon = 1.0) on Artificial Characters dataset (higher is better)
>
> | Model          | Test Accuracy | Test F1 Score |
> |----------------|---------------|---------------|
> | DP Naive Bayes | 0.231 | 0.176 |
> | DP Decision Tree | 0.246 | 0.138 |
> | DP GBDT [1]    | 0.224 | 0.210|
> | Tab-PE + Decision Tree | 0.403 | 0.394 |
> | Tab-PE + TabICL|0.504 | 0.500|
> | Tab-PE + TabPFN|0.511 | 0.501|
>
> It is worth noting that for low-order datasets, the direct DP training methods still outperform classifiers trained on synthetic data generation. In particular, DP Gradient Boosted Decision Tree (GBDT) achieves 83.61% and 94.19% accuracy on Adult and Breast Cancer, compared to the best classifiers trained on synthetic data at 83.31% and 92.17%. This result indicates that **the existing SOTA methods for direct DP classification training performs well for low-order datasets but struggle to capture high-order correlations**.
>
> &nbsp;
>
> **2. Interpretability**: Preserving high-order correlations leads to better performance for interpretability tasks. The following table presents the feature importance calculated using SHAP values [2] of classifiers trained on synthetic data.
>
> Table 2. SHAP Feature Importance for Class 0 (values closer to the ones by private data are better)
> | Dataset: Artificial Characters | Feature 1 | Feature 2 | Feature 3 | Feature 4 | Feature 5 | Feature 6 | Feature 7 |
> |-----------------------------|----|----|----|----|----|----|----|
> | Private Data                | 0.559 | 0.191 | 0.588 | 0.366 | 0.473 | 0.139 | 0.404 |
> | PrivGSD (best accuracy baseline)     | 0.126 | 0.099 | 0.093 | 0.157 | 0.153 | 0.265 | 0.204 |
> | AIM | 0.023 | 0.010 | 0.003 | 0.014 | 0.006 | **0.079** | 0.148 |
> | Tab-PE                      | **0.626** | **0.279** | **0.473** | **0.415** | **0.429** | 0.517 | **0.513** |
>
> &nbsp;
>
> Table 3. SHAP Feature Importance for Class 1
> | Dataset: Artificial Characters | Feature 1 | Feature 2 | Feature 3 | Feature 4 | Feature 5 | Feature 6 | Feature 7 |
> |-----------------------------|----|----|----|----|----|----|----|
> | Private Data                | 0.523 | 0.244 | 0.992 | 0.353 | 0.180 | 0.347 | 0.946 |
> | PrivGSD (best accuracy baseline)     | **0.376** | 0.035 | 0.356 | 0.158 | **0.204** | 0.165 | 0.463 |
> | AIM | 0.090 | 0.005 | 0.005 | 0.005 | 0.006 | 0.031 | 0.277 |
> | Tab-PE                      | 0.763 | **0.127** | **0.622** | **0.540** | 0.487 | **0.326** | **0.899** |
>
> &nbsp;
>
> **3. Data Imputation**: Tab-PE offers better predictive performance not only for the label column but also for missing feature values. The following table presents the imputation accuracy using models trained on synthetic data by different methods.
>
> Table 4. Feature Imputation Accuracy (higher is better)
> | Dataset: Artificial Characters | Overall | Feature 1 | Feature 2 | Feature 3 | Feature 4 | Feature 5 | Feature 6 | Feature 7 |
> |----------------|-----------|-----------|-----------|-----------|-----------|-----------|-----------|-----------|
> | PrivGSD (best accuracy baseline)    | 0.449 | **0.487** | 0.686 | 0.489 | 0.547 | 0.405 | 0.376 | 0.266 |
> | AIM | 0.335 | 0.466 | 0.617 | 0.354 | 0.433 | 0.142 | 0.271 | 0.201 |
> | Tab-PE | **0.495** | 0.481 | **0.700** | **0.562** | **0.568** | **0.472** | **0.416** | **0.317** |
>
> ---
>
> [1] Maddock, Samuel et al. "Federated Boosted Decision Trees with Differential Privacy." CCS, 2022
>
> [2] Scott M. Lundberg and Su-In Lee. A unified approach to interpreting model predictions. NeurIPS, 2017.

---

> ### Author Response · Authors · 2025-11-21
> **Response to Reviewer HPHj (2/2)**
>
> ### **Weakness 2 & Question 2: High-order vs Low-order Settings**
> We thank the reviewer for pointing out the missing highlights in Table 5. We confirm the numbers in Table 5 are correct. We will add the highlights in the revised version. It is worth noting that marginal-based methods such as AIM remain strong baselines. A recent benchmark [1] shows that AIM is still the SOTA on **both** fidelity metrics or ML utility. Outperforming the SOTA method on any metric is nontrivial. Moreover, Tab-PE is designed to preserve high-order interactions, which are not well captured by TVD fidelity metrics. As a result, the advantages of Tab-PE are primarily revealed by ML utility and high-order–sensitive embedding metrics, rather than TVD fidelity metrics.
>
> Here we give an example to show that **the TVD metrics do not reveal the preserved high-order correlation**. Let us consider the stress test (XOR) dataset with three features. In this simulation dataset, there is only one *single* correlation injected: the dependence of label on the features. Achieving strong 1-TVD fidelity requires the features in the synthetic data to be distributed similarly to the private data (uniform in this case). However, this does not guarantee that the labels are assigned correctly according to the underlying high-order dependency.
>
> Because the underlying function is known, so we can directly compute the raw probability that a sample's label is correctly assigned given the feature values:
>
> $$
> P(Y=1 \mid (X_1>0 \land X_2<0 \land X_3<0)
>         \cup (X_1<0 \land X_2>0 \land X_3<0)
>         \cup \dots )
> $$
>
> The following table presents the TVD fidelity metrics of AIM and Tab-PE. Although Tab-PE has higher TVD errors, 91% of Tab-PE samples have the correct label, reflecting that it preserves the true high-order interaction. In contrast, AIM fails to capture this dependency. Although we have configured the desired degree hyperparameter (i.e., 4) for AIM, its dynamic query selection mechanism struggles to identify the right high-order marginals under DP noise and the large search space of 1–4-way queries.
>
> | Method  | 1-TVD  | 2-TVD  | Prediction Acc | $P(Y=1 \| X1,X2,X3)$ |
> |----------|--------|--------|----------------|-------------------|
> | AIM  | 0.006  | 0.035  | 0.510         | 0.502             |
> | Tab-PE   | 0.036  | 0.143  | 0.940          | 0.910             |
>
> Finally, the raw probability naturally aligns with downstream ML utility which highlights that the downstream performance is a practical and reliable indicator of preserved high-order correlations.
>
> &nbsp;
>
> ### **Weakness 3 & Question 3: Public information**
>
> Regarding the class distribution, we have already provided an analysis in Appendix C.8 when the class distribution is not public, we spend a bit privacy budget to compute the private distribution. The performance of Tab-PE does not change significantly.
>
> Regarding the domain extraction, in our implementation, Tab-PE and all baselines have access to the same public domain information, including the domain sizes of categorical features and the bounds of numerical features for preprocessing. The results in the paper were reported under this **fair setting**.
>
> Here we provide additional results when the numerical bounds are 5x larger than the actual ranges. The performance of Tab-PE is fairly robust even with the large errors of bounds and significantly outperforms the best baseline using public feature bounds on ML ultilities (PrivGSD's test accuracy: ~40%).
>
> | Setting | Test Acc | Test F1 | 1-TVD | 2-TVD |
> |----------------|-----------|-----------|-----------|-----------|
> | Tab-PE (true bounds) | 0.504 | 0.500 | 0.183 | 0.380 |
> | Tab-PE (5x bounds) | 0.485 | 0.474 | 0.197 | 0.389 |
>
> &nbsp;
>
> ### **Question 1: Potential Impacts**
>
> We appreciate the reviewer for asking about our claim.
>
> By new paradigm, we refer not to large gains on all benchmarks, but to shift the focus from modeling low-order marginals to high-order interactions. While the SOTA methods have been dominated by query-based approaches for years, Tab-PE offers a new direction without using statistical queries to enable modeling capacity and efficiency that the prior methods may struggle to achieve. We believe this new high-order perspective and the proposed approach can inspire new research directions in the field. Additionally, we will clarify this in the revised version to avoid confusion and misinterpretation.
>
> ---
> [1] Benchmarking Differentially Private Tabular Data Synthesis, K Chen, X Li, C Gong, R McKenna, T Wang, SIGMOD 2026

---

> > ### Comment · Reviewer_HPHj · 2025-11-21
> >
> > I appreciate the efforts made to clarify the method and provide additional experiments that demonstrate the practical use and privacy analysis of attribute information. However, I still have concerns about the performance improvements and the claim of modeling higher-order correlations.
> >
> > * Previous statistic-based methods explicitly optimize the low-order marginals, but this is a methodological choice, not an explanation for why Tab-PE would underperform. Different synthesis algorithms can optimize various aspects of the tabular data, but they must all be evaluated using the same criteria. At least for low-order marginals, Tab-PE does not show an advantage.
> >
> > * I do not believe that the XOR stress test could rigorously demonstrate the advantages of Tab-PE's higher-order preservation (although it's good to have it to motivate the paper). Specifically, Tab-PE uses label conditions to generate synthetic data, so it naturally performs better in preserving the correlation between labels and other attributes. To the best of my knowledge, these baselines do not rely on this for data generation. Thus, this does not mean it can really preserve better higher-order marginals. A more straightforward (or even simpler) approach would be to use higher-order marginal distribution similarities (e.g., 4 or 5 marginal distribution similarities using Wasserstein distance) on real-world datasets. It's questionable whether Tab-PE will outperform the baselines in this setting, especially since it already struggles with preserving similarity in low-order marginals.
> >
> > * Given these concerns, I do not think that Tab-PE represents a breakthrough in differential privacy for tabular data synthesis. While it can be a new approach for synthesis, the core goal of differential privacy (DP) data synthesis is to perform arbitrary tasks without consuming additional privacy budgets. For tabular data, one of the key tasks is preserving marginal distributions (whether low or high-order), which is crucial for downstream tasks like range queries. If the goal is only classification, one should directly train a DP classifier rather than generate synthetic data. More evidence is needed (though I am unsure whether all of this can be addressed in the rebuttal) to claim that this approach represents a new paradigm for the field.

---

> > > ### Author Response · Authors · 2025-11-27
> > > **Follow-Up Response to Reviewer HPHj (1/3)**
> > >
> > > We would like to thank the reviewer for their constructive and insightful suggestions.
> > >
> > > &nbsp;
> > >
> > > ## **Unconditional Generation**
> > > Tab-PE can also perform unconditional generation, where the label can be consider as a feature. The ML ultility performance does not change much, while the TVD is slightly worse. For unconditional generation, Tab-PE requires more number of iterations than conditional generation, (20 iterations vs 15 iterations).
> > >
> > > &nbsp;
> > > | Method | Val Accuracy | Macro F1    | Avg 1-TVD | Avg 2-TVD |
> > > |--------|--------------|-------------|-----------|-----------|
> > > |Tab-PE (conditional) | 0.5003261579 | 0.4960592772 | 0.183     | 0.380     |
> > > |Tab-PE (unconditional)| 0.5003261579 | 0.4962993042 | 0.200     | 0.386     |
> > >
> > > &nbsp;
> > >
> > > ## **Analysis on TVD, JSD, Wasserstein distance**
> > > Thank you again for the great suggestions, here to understand more about the high-order fidelity metrics, we conduct some analysis about TVD, JSD (Jensen-Shannon Divergence), and Wasserstein fidelity metrics at different orders and settings.
> > >
> > > For $k$-TVD and $k$-JSD, we first discretize the data into bins. Then, for each $k$-column combination, we measure the empirical joint frequencies in both private and synthetic datasets and convert them into probability mass functions by normalizing the counts. Subsequently, we compute the TVD or JSD between the joint distributions of the two datasets. We repeat the process for all $k$-column combinations and calculate the average.
> > >
> > > For the Wasserstein distance, all the features are scaled to [0, 1], so no feature dominates the distance calculation. The Wasserstein distance is computed directly on the full $k$-dimensional subsets for each $k$-column combination. The $k$-W is the average value of all combinations. The pairwise cost matrix is built using the standard Euclidean (l2) distance.
> > >
> > > Please refer to the tables for the detailed results. We summarize the key observations as follows:
> > > - The choice of discretization bins can impact to TVD and JSD significantly, especially on low-order ones, depending on the dataset. Wasserstein distance does not require discretization.
> > > - TVD and JSD are biased toward the synthetic data size. In the paper, AIM's TVD is measured with the private data size, while the size of Tab-PE synthetic data is generally smaller. This makes the TVD results in the paper favor AIM, creating a large gap between Tab-PE and AIM.
> > > - Wasserstein distance is more robust to the synthetic data size.
> > > - The general trend is that AIM performs better on lower-order fidelity metrics, while Tab-PE performs better on higher-order fidelity metrics.
> > > - With the same size of dataset, the low-order fidelity metrics gap between AIM and Tab-PE reduces significantly.

---

> > > ### Author Response · Authors · 2025-11-27
> > > **Follow-Up Response to Reviewer HPHj (2/3)**
> > >
> > > ### **Artificial-Characters Dataset**
> > >
> > > &nbsp;
> > >
> > > | **10 bins**  | 1-TVD  | 2-TVD  | 3-TVD | 4-TVD | 5-TVD | 6-TVD | 7-TVD | 8-TVD |
> > > |----------|--------|--------|-------|-------|-------|-------|-------|-------|
> > > | AIM (private size) | **0.026881** | **0.137579** | **0.311344** | **0.535431** | 0.746572 | 0.882745 | 0.950171 | 0.979586|
> > > | AIM (Tab-PE's size) | 0.067188 | 0.238479 | 0.534491 | 0.801122 | 0.928104 | 0.975327 | 0.992710 | 0.998182 |
> > > | Tab-PE | 0.075617 | 0.183075 | 0.352743 | 0.553896 | **0.712036** | **0.812589** | **0.873387** | **0.911038** |
> > > | **20 bins** |
> > > | AIM (private size)        | **0.035445** | **0.176709** | **0.417548** | **0.698039** | **0.876618** | **0.954169** | 0.984497 | 0.995805 |
> > > | AIM (Tab-PE's size) | 0.067188 | 0.238479 | 0.534491 | 0.801122 | 0.928104 | 0.975327 | 0.992710 | 0.998182 |
> > > | Tab-PE | 0.182792 | 0.379879 | 0.638086 | 0.832459 | 0.920391 | 0.957522 | **0.975403** | **0.984717**|
> > >
> > > &nbsp;
> > >
> > > | **10 bins**  | 1-JSD  | 2-JSD  | 3-JSD | 4-JSD | 5-JSD | 6-JSD | 7-JSD | 8-JSD |
> > > |----------|--------|--------|-------|-------|-------|-------|-------|-------|
> > > | AIM (private size) | **0.002002** | **0.028840** | **0.110675** | **0.262665** | 0.440088 | 0.569201 | 0.637855 | 0.670122|
> > > | AIM (Tab-PE's size) | 0.004210 | 0.041247 | 0.162235 | 0.369182 | 0.540581 | 0.629366 | 0.667776 | 0.684533 |
> > > | Tab-PE | 0.009173 | 0.041745 | 0.136954 | 0.288252 | **0.423094** | **0.514129** | **0.570781** | **0.607037** |
> > > | **20 bins** |
> > > | AIM (private size) | **0.004184** | **0.047223** | **0.192978** | **0.421651** | **0.578038** | **0.648126** | **0.676540** | 0.688806 |
> > > | AIM (Tab-PE's size) | 0.006742 | 0.071385 | 0.281982 | 0.510184 | 0.625681 | 0.669483 | 0.686010 | 0.691374 |
> > > | Tab-PE | 0.048714 | 0.148365 | 0.363162 | 0.539298 | 0.620076 | 0.655000 | 0.672700 | **0.683023** |
> > >
> > > &nbsp;
> > >
> > > | Wasserstein  | 1-W  | 2-W  | 3-W | 4-W | 5-W | 6-W | 7-W | 8-W |
> > > |----------|--------|--------|-------|-------|-------|-------|-------|-------|
> > > | AIM (private size) | **0.009605** | **0.037011** | **0.078604** | 0.131570 | 0.188566 | 0.245150 | 0.299776 | 0.351410 |
> > > | AIM (Tab-PE's size) | 0.014817 | 0.044442 | 0.087082 | 0.140160 | 0.197000 | 0.253496 | 0.307995 | 0.359972 |
> > > | Tab-PE | 0.037793 | 0.065208 | 0.095395 | **0.130072** | **0.164409** | **0.196297** | **0.225520** | **0.252635** |

---

> > > ### Author Response · Authors · 2025-11-27
> > > **Follow-Up Response to Reviewer HPHj (3/3)**
> > >
> > > ### **Person-Activity Dataset**
> > > Due to the computational cost, as this dataset is signficantly larger than Artificial-Characters, we only present the results of the same size synthetic data size.
> > >
> > > &nbsp;
> > >
> > > | **10 bins**  | 1-TVD  | 2-TVD  | 3-TVD | 4-TVD | 5-TVD | 6-TVD | 7-TVD | 8-TVD |
> > > |----------|--------|--------|-------|-------|-------|-------|-------|-------|
> > > | AIM (Tab-PE's size) | **0.012771** | **0.053379** | **0.143229** | **0.296869** | **0.488874** | 0.668335 | 0.809941 | 0.900310 |
> > > | PE      | 0.038448 | 0.090559 | 0.188484 | 0.345468 | 0.514707 | **0.652367** | **0.750849** | **0.814606** |
> > > | **20 bins**  |
> > > | AIM (Tab-PE's size) | **0.017450** | **0.094118** | **0.273800** | **0.536738** | **0.761298** | 0.897294 | 0.964074 | 0.989780 |
> > > | PE      | 0.044399 | 0.132814 | 0.324634 | 0.573495 | 0.762776 | **0.868473** | **0.922113** | **0.949755** |
> > >
> > > &nbsp;
> > >
> > > | **10 bins** | 1-JSD    | 2-JSD    | 3-JSD    | 4-JSD    | 5-JSD    | 6-JSD    | 7-JSD    | 8-JSD    |
> > > |---------|----------|----------|----------|----------|----------|----------|----------|----------|
> > > | AIM (Tab-PE's size)| **0.000177** | **0.005092** | **0.027778** | **0.105727** | **0.237561** | 0.378928 | 0.500337 | 0.585041 |
> > > | PE      | 0.001500 | 0.012480 | 0.048344 | 0.138716 | 0.261444 | **0.372463** | **0.455126** | **0.510942** |
> > > | **20 bins**  |
> > > | AIM (Tab-PE's size)| **0.000315** | **0.013355** | **0.093527** | **0.273425** | **0.457828**     | 0.582623     | 0.649735     | 0.678809     |
> > > | PE      | 0.002247 | 0.024329 | 0.121237 | 0.299010 | 0.459269 | **0.558621** | **0.611397** | **0.639753** |
> > >
> > >
> > > &nbsp;
> > >
> > > | Wasserstein  | 1-W  | 2-W  | 3-W | 4-W | 5-W | 6-W | 7-W | 8-W |
> > > |----------|--------|--------|-------|-------|-------|-------|-------|-------|
> > > | AIM (Tab-PE's size)| **0.006365** | **0.015632** | **0.034485** | **0.062834** | 0.096774 | 0.133273 | 0.171545 | 0.210860 |
> > > | Tab-PE | 0.006947 | 0.016937 | 0.035587 | 0.062057 | **0.091398** | **0.120872** | **0.149365** |  **0.177133** |
> > >
> > >
> > > &nbsp;
> > >
> > > &nbsp;
> > >
> > > ### **(Simulation dataset) SCN-NN**
> > >
> > > &nbsp;
> > >
> > > | Wasserstein  | 1-W  | 2-W  | 3-W | 4-W | 5-W | 6-W | 7-W | 8-W |
> > > |----------|--------|--------|-------|-------|-------|-------|-------|-------|
> > > | AIM (Tab-PE's size)| 0.019650 | 0.034025 | 0.055341 | 0.084672 | 0.118352 | 0.153289 | 0.187920 | 0.221675 |
> > > | Tab-PE | **0.017594** | **0.032290** | **0.054305** | **0.084385** | **0.117958** | **0.151657** | **0.184001** | **0.214603** |
> > >
> > > &nbsp;
> > >
> > > ---
> > > ---
> > >
> > > &nbsp;
> > >
> > > We greatly appreciate the reviewer's time and thoughtful suggestions. If any new questions arise, we are happy to provide further clarification.
> > >
> > > &nbsp;
> > >
> > > Best regards,
> > >
> > > The authors

---

### Official Review · Reviewer_NVau · 2025-11-11

**Soundness:** 4
**Presentation:** 4
**Contribution:** 3
**Rating:** 8
**Confidence:** 4

**Summary:**

This paper introduced Tab-PE, a proposed method extending the private evolution (PE) framework for text and images into tabular data. Unlike prior PE methods relying on foundation models, Tab-PE employs simple heuristic APIs to evolve candidate synthetic samples through controlled perturbations and DP nearest-neighbor scoring. The paper argues that this approach better models high-order correlations in tabular data where marginal-based methods struggle.

**Strengths:**

- The paper is well written and easy to follow. It highlights its novel contributions well and makes clear and distinct references to previous literature. Figures are clear and well motivated.
- The adaptation of the PE framework to tabular data is novel and interesting, and has not been studied in the literature before.
- The paper makes an important empirical point that current DP tabular synthesis methods implicitly optimize for low-order correlations and fail under high-order dependencies. The “XOR stress test” is a clear and well-designed diagnostic benchmark.
- Tab-PE achieves consistent improvements across varied datasets and privacy levels, showing both utility gains and significant runtime savings. The scalability of the method makes it highly useful for real-world applications, such as financial or healthcare tabular data. Extensive experiments across synthetic (XOR, SCM) and real-world datasets demonstrate substantial performance gains (up to +10% accuracy) and improved computational efficiency (up to 28× faster than AIM) under comparable privacy budgets.
- The baselines are well chosen and ablations study is well-done with the selection strategy, polynomial decay schedule, and hyperparameter sensitivity studies and strengthen confidence in the design choices.

**Weaknesses:**

- While the authors reuse DP composition results from prior PE papers, the paper does not formalize why the DP nearest-neighbor histogram implicitly captures high-order correlations, as many other DP papers do. A more rigorous connection between the algorithmic process and statistical estimation of joint distributions would improve the conceptual depth. A theoretical contribution on how the nearest-neighbor scoring function relates to statistical query families or mutual information under DP noise would be helpful to make this paper useful beyond a purely engineering/empirical perspective.

**Questions:**

- Does Tab-PE’s advantage persist under stricter privacy (ε < 1)? Some prior DP synthesizers degrade sharply; how robust is your method in low-privacy regimes?
- For methods like AIM and RAP++, were hyperparameters re-tuned under identical privacy budgets, or were default settings used? Could differences in optimization explain part of the observed utility gap?
- How do you choose the neighborhood radius or kernel bandwidth in the DP_NN_HISTOGRAM? Is it tuned on public data, or does it affect privacy accounting if selected adaptively?

---

> ### Author Response · Authors · 2025-11-21
> **Response to Reviewer NVau**
>
> We thank the reviewer for the thoughtful feedback and constructive comments. We response the main concerns as follows:
> ### **Weakness 1: Theoretical Justification**
> This is a great suggestion. Although Tab-PE has been empirically demonstrated to be effective, we acknowledge that a theoretical understanding of the algorithm would strengthen the work. We will explore this perspective.
>
> &nbsp;
>
> ### **Question 1: Strict privacy ($\epsilon < 1$)**
> Thank you for mentioning the strict privacy regime. As shown in Figures 3 and 4, Tab-PE consistently outperforms the baselines across datasets and small privacy budgets ($\epsilon=0.5$ and $\epsilon=0.2$). This indicates the robustness of Tab-PE in the strict privacy regimes.
>
> &nbsp;
>
> ### **Question 2: Baseline Hyperparameters**
> We reuse the hyperparameters from a recent unified benchmark [1]. As our focus on the high-order correlations, we tune the order of marginal queries in all baselines as this hyperparameter is important to enable the baselines to capture high-order correlations. We reported the best results across the query orders from 2 to 5. In the stress test experiment, we also configure the desired order to match with the underlying correlation order to maximize the baselines' success.
>
> &nbsp;
>
> ### **Question 3: DP_NN_HISTOGRAM**
> We thank the reviewer for bring this point. Our current DP_NN_HISTOGRAM is a simple but effective neareast neighbor-based assignment. Our current implementation does not employ a constrain of neighborhood radius. While adding this constrain can avoid cases that a synthetic sample is voted but actually far away, it also creates a risk of early suboptimal convergence as the algorithm does not enhance the exploration ability. By not constraining, we can ensure that the evolution can explore more diverse candidates before refinement. We will add this discussion for potential future work.
>
> ---
>
> [1] Benchmarking Differentially Private Tabular Data Synthesis, K Chen, X Li, C Gong, R McKenna, T Wang, SIGMOD 2026

---

### Author Response · Authors · 2025-12-04
**Author Remarks**

Dear ACs, SACs, and Reviewers,

We sincerely thank the reviewers, ACs, and SACs for their time and valuable feedback on our submission. We are grateful for the comments and constructive suggestions. Below, we provide a brief summary of our contributions and the main concerns raised by reviewers, and how we have addressed them.

&nbsp;

## **Key Contributions:**
1. **Insights** We examine widely used benchmarking datasets and show that they are insufficient to reveal whether synthesizing methods can capture high-order interactions. This helps explain the limited progress in the field: a method from January 2022 remains state-of-the-art and *all* leading methods employ the same strategy, optimizing for low-order queries.
2. **New Benchmark** We introduce a new benchmark that contains high-order correlations and show that existing methods struggle to model them.
3. **New Method** We propose Tab-PE, a novel method based on Private Evolution that can effectively capture high-order correlations while running faster up to 28x faster than the prior state-of-the-art techniques.


&nbsp;

## **Strengths Highlighted by Reviewers:**
- **Novelty** Across reviews, there are recognitions that adapting the Private Evolution framework to tabular data without relying on LLMs is a novel and interesting direction (Reviewer #NVau) and conceptually straightforward yet very effective (Reviewer #QPTQ).

- **Clarity and Presentation** Reviewer #NVau, #HPHj, and #QPTQ all agreed that paper is clear, easy to follow, and well-organized.

- **Low- vs High-Order Correlation Analysis** Reviewer #Nvau commented the contribution in analyzing existing benchmarks and high-order vs low-order correlations, and together with Reviewer #HPHj highlighted the proposed XOR stress test is motivated and intuitive to illustrate weaknesses of the marginal-based methods. Reviewer #QPTQ further observed that our paper clearly identifies and empirically validates a meaningful limitation in existing DP-SDG algorithms in capturing high-order correlations.

- **Algorithms** Reviewer #HPHj and #9tC8 emphasized that Tab-PE is simple, easy to implement, and computationally efficient. Reviewer #9tC8 also noted that it requires no model training and scales efficiently to medium-sized tabular datasets, offering a practical alternative to the extensively studied query- or model-based approaches.

&nbsp;

## **Addressing Reviewers' Concerns:**
We highlight the main concerns raised by the reviewers, some minor issues seem to come from points that may have been overlooked. We referred to the specific sections of the original submission where these points were already addressed.

- **Setting Novelty**: Reviewer #9tC8 cited two prior works: SoK and Kamino. We clarified the distinction between our settings and constrain-aware synthesis (Kamino). The SoK paper mentions the limitations of the existing methods using low-order queries and considers high-order correlations as **an open problem**. Our work provides the first empirical demonstration of these limitations and of why widely used benchmarks overlook them.

- **Method Novelty** (Reviewer #QPTQ): We clarified the key differences between Tab-PE and both PrivGSD and Swanberg et al. Tab-PE does not rely on LLMs or statistical queries. that enables Tab-PE to be significantly more efficient than the prior works.

- **Public Information Assumption** (Reviewer #HPHj): We clarified that the public information used follows common practice and all baselines in our experiments rely on the same information for a fair comparison.

- **High-order correlation formalization** (Reviewer #9tC8): We provided a formal definition via the lens of mutual information.

- **Practical Usefulness of high-order correlations** (Reviewer #HPHj): We added 3 applications including Classification (Tab-PE also outperforms direct DP training classifiers), Interpretability (via SHAP values), and Data Imputation.

- **Lower performance on low-order marginals but better capturing high-order correlations** (Reviewer #HPHj and #9tC8): We first explained why low-order metrics (1-TVD and 2-TVD) do not reveal performance  on high-order interactions, using a concrete example where AIM achieves strong low-order fidelity yet fails entirely to capture a 4-way dependency. After receiving the reviewers' responses, we extended our analysis with additional TVD, JSD, and Wasserstein evaluations. The results show that TVD and JSD are biased towards the number of synthetic samples, this makes the 1-TVD and 2-TVD results presented in our original tables favor AIM. With the same synthetic sample size, Tab-PE performs comparably on low-order marginals while better on high-order metrics. Unfortunately, these extended results were added after the leakage incident, the reviewers may not have had the opportunity to see them.
---

&nbsp;

We greatly appreciate the time and effort of the reviewers, ACs, and SACs in evaluating our submission.

*The Authors*

---

### Meta-Review · Area_Chair_mxk7 · 2026-01-04

**Summary:**

This paper applies the private evolution algorithm to differentially private tabular data generation and shows PE is effective for high order correlation. This paper received a diverse set of scores (2, 4, 6, 8).

Reviewers acknowledge the motivation, appreciate the simplicity of the algorithm, and like the XOR illustrative examples. However, reviewers also raised concerns on lack of theoretical depth on higher order correlation, limited algorithmic novelty compared to the original PE algorithm, proper discussion of closely related work (Swanberg et al., PrivGSD, Hu et al.), fair comparison with other tabular data method on existing benchmark, and notions and general claims.

Some of the concerns are inherently hard to address, but not necessarily the key factor for paper decisions (e.g., lack of theoretical depth, limited algorithmic novelty); the clarification questions on related work are reasonably responded, but unfortunately did not seem to lead to score change due to coupling to the contribution concerns. Additional experiments are provided that partially addressed the comparison concerns.

In general, this paper has some interesting points that are valuable to the community. However, I also agree with reviewers on notions and claims. While I understand the intention of using “Differentially Private Synthetic Data via APIs 4” to draw connections to previous PE methods, “APIs” seems to be unnecessary and even confusing; “4” also seems to be unnecessary and unconventional in a double blind peer reviewed venue. Also considering the reviewers’ concerns on inaccurate claims and the impact in the community, I have to unfortunately use a high bar and borderline review comments with two negative reviews (reviewer HPHj comments on maintaining score 4, and reviewer 9tC8 comment on remaining concerns that unlikely to flip score 2 to positive) cannot warrant acceptance.

**Reviewer Concerns:**

Reviewers acknowledge the motivation, appreciate the simplicity of the algorithm, and like the XOR illustrative examples. However, reviewers also raised concerns on lack of theoretical depth on higher order correlation, limited algorithmic novelty compared to the original PE algorithm, proper discussion of closely related work (Swanberg et al., PrivGSD, Hu et al.), fair comparison with other tabular data method on existing benchmark, and notions and general claims.

Some of the concerns are inherently hard to address, but not necessarily the key factor for paper decisions (e.g., lack of theoretical depth, limited algorithmic novelty); the clarification questions on related work are reasonably responded, but unfortunately did not seem to lead to score change due to coupling to the contribution concerns. Additional experiments are provided that partially addressed the comparison concerns.

**Reviewer Scores:**

This paper received a diverse set of scores (2, 4, 6, 8). Reviewers did not mention score changes in discussion, and unfortunately, two negative reviews are likely to maintain (reviewer HPHj comments on maintaining score 4, and reviewer 9tC8 comment on remaining concerns that unlikely to flip score 2 to positive).

---

### Decision · Program_Chairs · 2026-01-26

Reject